# A fundamental viewpoint on the hydrogen spillover phenomenon of electrocatalytic hydrogen evolution

Jiayuan Li[1], Jun Hu[2 ✉], Mingkai Zhang[3], Wangyan Gou[1], Sai Zhang[1], Zhong Chen [4], Yongquan Qu [1,3 ✉] & Yuanyuan Ma[1 ✉]

Hydrogen spillover phenomenon of metal-supported electrocatalysts can significantly impact their activity in hydrogen evolution reaction (HER). However, design of active electrocatalysts faces grand challenges due to the insufficient understandings on how to overcome this thermodynamically and kinetically adverse process. Here we theoretically profile that the interfacial charge accumulation induces by the large work function difference between metal and support ($\Delta\Phi$) and sequentially strong interfacial proton adsorption construct a high energy barrier for hydrogen transfer. Theoretical simulations and control experiments rationalize that small $\Delta\Phi$ induces interfacial charge dilution and relocation, thereby weakening interfacial proton adsorption and enabling efficient hydrogen spillover for HER. Experimentally, a series of Pt alloys-CoP catalysts with tailorable $\Delta\Phi$ show a strong $\Delta\Phi$-dependent HER activity, in which PtIr/CoP with the smallest $\Delta\Phi = 0.02$ eV delivers the best HER performance. These findings have conclusively identified $\Delta\Phi$ as the criterion in guiding the design of hydrogen spillover-based binary HER electrocatalysts.

[1] Key Laboratory of Special Functional and Smart Polymer Materials of Ministry of Industry and Information Technology, School of Chemistry and Chemical Engineering, Northwestern Polytechnical University, Xi'an, China. [2] School of Chemical Engineering, Northwest University, Xi'an, China. [3] Frontier Institute of Science and Technology, Xi'an Jiaotong University, Xi'an, China. [4] School of Materials Science and Engineering, Nanyang Technological University, Singapore, Singapore. ✉email: hujun32456@163.com; yongquan@xjtu.edu.cn; yyma@nwpu.edu.cn

Hydrogen, featured by high gravimetric energy density and zero-emission, holds great potentials as the sustainable alternatives to fossil fuels[1,2]. Its promising application prospect has stimulated the intensive investigations to develop various methodologies for the scalable and cost-effective production of hydrogen[3]. Electrocatalytic water hydrolysis offers an attractive approach to generate hydrogen with high purity[4]. However, this technology is still far from being commercially competitive to the hydrogen production through steam reforming. The key issue is to look for robust electrocatalysts with maximized catalytic efficiency and high longevity under the operating conditions, especially for those in acidic electrolytes[5,6].

Currently, catalyst design for electrocatalytic hydrogen evolution reaction (HER) is guided by the classic volcano theory in which hydrogen adsorption free energy ($\Delta G_H$) on catalyst surface is used as an indicator of the catalytic efficiency[7]. In this theory, neither too strong nor too weak adsorption of active hydrogen species ($\Delta G_H \approx 0$) on catalyst surface is recognized as the criterion for efficient HER electrocatalysts, since this state can facilitate both hydrogen adsorption and desorption[7]. Therefore, the design of electrocatalysts to be as closer as possible to the "volcano" top is of intensive academic and industrial interest[7,8]. With this concept in mind, platinum (Pt)-based materials are advocated as the best candidates due to their near ideal $\Delta G_H$[9,10]. Considering the scarcity of Pt in nature, various earth-abundant transition metal compounds (e.g., sulfides, phosphides, carbides, and nitrides) as the alternatives have been explored[11–16]. Although the significant progress has been made so far, the state-of-the-art inexpensive HER electrocatalysts still suffer from relatively low activity and poor durability. Especially in harsh acidic media, the deactivation is often observed under this corrosive conditions[17,18]. Thus, it calls for further exploitation of highly efficient, robust, and cost-effective HER electrocatalysts through conceptual innovation.

Inspired by the hydrogen spillover phenomenon in thermal hydrogenation through the strong metal-support interaction[19,20], hydrogen spillover has recently emerged as a frontier in the binary metal/support HER electrocatalysts[21–26], which undergoes (1) strong proton adsorption on metals ($\Delta G_{H\text{-metal}} < 0$); (2) interfacial hydrogen spillover from metals to supports and (3) efficient hydrogen desorption on supports ($\Delta G_{H\text{-support}} > 0$). This strategy affords a concept by integrating both advantages of metal and support and thereby kinetically promotes the proton adsorption and hydrogen desorption. Also, this design conception is cost-effective as it can reduce the metal usage while still deliver the competitive catalytic performance[22,24]. Unfortunately, the successful cases based on this conception are still rare due to the lack of the fundamental understandings on what are the key factors behind the hydrogen spillover process, a thermodynamically and kinetically unfavorable process for HER. Thus, the fundamental understandings on how to enable energetically favorable hydrogen spillover under HER operating conditions are crucial to the design and synthesis of such hydrogen spillover-based binary (HSBB) catalysts.

In this work, by theoretically profiling the hydrogen spillover kinetics in such HSBB catalysts, the occurrence of the interfacial hydrogen spillover is predicted to be determined by the work function difference between metal and support ($\Delta\Phi$). Large value of $\Delta\Phi$ leads to interfacial charge accumulation, strong proton trapping at the interface, and thereby unfavorable hydrogen spillover kinetics from metal to support during HER. Inspired by this finding, the modulation of work function of metal by alloying with a second metal to match that of the support provides a practical strategy to minimize the interfacial charge accumulation and enable efficient the interfacial hydrogen transfer. To examine our theoretical predictions, a suite of alloyed Pt nanoparticles

anchored on CoP (PtM/CoP, M = Rh, Pd, Ag, Ir, and Au) were prepared as model catalysts to investigate the hydrogen spillover kinetics for HER, where the $\Delta\Phi$ was used as the criterion. We experimentally and theoretically demonstrated effective hydrogen spillover in PtIr/CoP due to their close work functions. The optimized $Pt_2Ir_1$/CoP with a low metal loading of $1.0\,wt\%$ delivered superior HER performance with a small Tafel slope of $25.2\,mV\cdot dec^{-1}$, low overpotentials of 7 mV at $20\,mA\cdot cm^{-2}$ and nearly undecayed stability over a period of 500 h. Our finding not only provides deep insights on the hydrogen spillover phenomenon in HER electrocatalysts, but also offers a promising strategy to tackle this fundamental challenge and establishes a thorough conceptive grounding for the design of highly performed HER electrocatalysts.

## Results

**Theoretical viewpoint on the hydrogen spillover phenomenon in binary HER electrocatalysts.** Generally, the interaction between a metal and a support with the large difference in their Fermi energies ($E_f$) constructs a Schottky junction at the interface (Fig. 1). Such a large difference in their $E_f$ drives the interfacial charge flow until the system reaches an equilibrium, followed by the Schottky barrier formation and charge accumulation at the interface of metal and support, thus endowing the interfacially catalytic sites with the strong proton absorption ability[27–29]. In this case, the interfacial hydrogen spillover from metal to support has to overcome a large energy barrier, leading to a kinetically difficult process and thereby unsatisfactory HER activity.

Conversely, when metal and support possess similar $E_f$, they should restrain the interfacial charge-transfer and minimize the interfacially accumulated charge (Fig. 1). With this electronic configuration, the energy barrier for the interfacial hydrogen spillover is expected to be significantly declined, theoretically resulting in a high catalytic HER activity. Inspired by the above rational analysis and the qualitative analysis for $E_f$ by the work function ($\Phi$) in materials[30], we hypothesize that the difference in $\Phi$ between metal and support ($\Delta\Phi = |\Phi_{metal} - \Phi_{support}|$) plays a crucial role in determining the transfer kinetics of the interfacial hydrogen spillover. The minimized $\Delta\Phi$ theoretically would reduce the hydrogen spillover barrier, resulting in efficient HER catalysis. If this hypothesis can be validated, the key issue for such

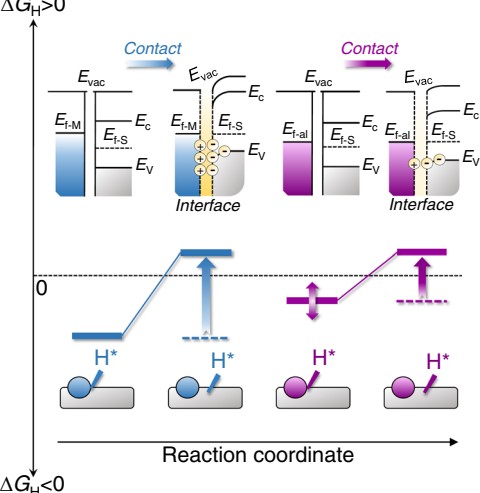

**Fig. 1 Schematic illustrations of the interfacial electronic configurations and hydrogen spillover phenomenon in HSBB catalysts.** $E_{vac}$ = vacuum energy, $E_c$ = conduction band, $E_v$ = valence band, $E_{f\text{-M}}$ = Fermi level of metal, $E_{f\text{-al}}$ Fermi level of alloy, $E_{f\text{-S}}$ Fermi level of support.

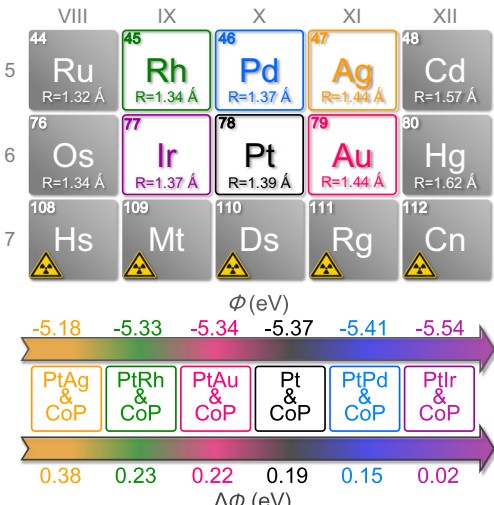

**Fig. 2 Catalyst Design.** Design of PtM/CoP model catalysts with the controllable $\Delta\Phi$.

binary catalysts mentioned above will be revealed accordingly and an enormous amount of time and expenses for catalyst design or selection will be saved through simply examining the $\Delta\Phi$ between the two components in the HSBB electrocatalysts. Different from the previous HER catalyst screening based on its energy at or near the lowest unoccupied states ($\varepsilon_{LUS}$)[31,32], the pivotal determinant of $\Delta\Phi$ herein is to tailor the energy barrier for the interfacial hydrogen spillover. Thereafter, the improved catalytic performance is realized by efficient hydrogen spillover to bridge the energetically favorable hydrogen adsorption/desorption on different active sites rather than by the modulated hydrogen adsorption/desorption on catalyst surface in previous reports[31,32].

**Rational design of PtM/CoP model catalysts.** Taking Pt/CoP as a model catalyst, the $\Delta\Phi$ of ~0.19 eV between Pt ($\Phi_{Pt}$ = 5.37 eV) and CoP ($\Phi_{CoP}$ = 5.56 eV) induces a large energy barrier for hydrogen spillover from Pt to CoP and thereby delivers no synergistic enhancements in the catalytic HER activity, consistent with our previous report[22]. The calculation details of work function could be found in "Methods" section and Supplementary Figs. 1, 2. According to the above analysis, the small $\Delta\Phi$ is expected to realize efficient hydrogen spillover, if $\Phi$ of either metal or support can be tailorable. Practically, the energy-level configuration of a metal can be precisely modulated by alloying with another metal[33,34]. Thus, alloying Pt is an effective strategy to regulate $\Delta\Phi$ between metal and CoP substrate.

Based on the principle that alloying between the neighbors in periodic table of elements is easy to happen without structural aberration, various foreign metals (M = Ir, Rh, Pd, Ag, and Au) with the period/group adjacent to Pt come into sight. Compared to that of CoP, the calculated values of $\Phi$ for PtIr, PtRh, PtPd, PtAg, and PtAu were 5.54 eV, 5.33 eV, 5.40 eV, 5.18 eV and 5.34 eV, respectively (Fig. 2 and Supplementary Fig. 1), indicating that PtIr/CoP might be the best candidate to achieve the kinetically favorable hydrogen spillover from metal to CoP and further comparing with other PtM/CoP model catalysts offer an opportunity to examine our viewpoint.

**Synthesis, characterizations, and catalytic performance.** Accordingly, Pt/CoP, Ir/CoP, PtAg/CoP, PtRh/CoP, PtAu/CoP, PtPd/CoP, and PtIr/CoP electrocatalysts were synthesized through the same method (details can be found in "Methods" section and Supplementary Figs. 3, 4). To confirm the successful

preparation, powder X-ray diffraction (XRD) curves of PtIr/CoP with a metal loading of 1.0 wt% and a Pt/Ir molar ratio of 2:1 (determined by inductively-coupled plasma mass spectrometry, ICP-MS, Supplementary Table 1) are shown in Fig. 3a as an example. XRD pattern is consistent with that of CoP standard (JCPDS #29-0497), suggesting unaltered phase of CoP during synthesis[35]. No XRD peaks of the loaded metal were observed due to the small size and low loading in the catalysts. The transmission electron microscopy (TEM) image (Fig. 3b) confirms the uniform distribution of the small PtIr alloyed metal nanoparticles of ~1.60 nm on CoP nanosheets. The magnified TEM image in Fig. 3c shows the interplanar spacings of 2.25 and 2.82 Å, corresponding to the respective (111) facet of Pt alloy with cubic lattice and the (011) facet of CoP with orthorhombic lattice. In the high-resolution X-ray photoelectron spectra (XPS) of PtIr/CoP (Supplementary Fig. 5), the typical signals of CoP, Pt and Ir species were identified[36–38]. Considering the binding energy of Pt $4f_{7/2}$ (71.0 eV) and Ir $4f_{7/2}$ (60.8 eV) in $Pt^0$ and $Ir^0$ benchmarks, a shift to higher binding energy for Pt $4f_{7/2}$ (71.2 eV) together with a shift to a lower value for Ir $4f_{7/2}$ (60.6 eV) in the case of PtIr/CoP were related to a lower electron density on the Pt due to the presence of Ir with the relatively higher electron affinity, suggesting the formation of a bimetallic PtIr alloy[37,39]. Energy-dispersive X-ray (EDX) mapping and line scan (Fig. 3c) for PtIr/CoP suggest uniform distribution of Co and P elements in nanosheets and concentrated distribution of Pt and Ir elements in nanoparticles, confirming the formation of PtIr/CoP hybrids. The carbon monoxide (CO) stripping voltammetry further verified the formation of the alloyed PtIr in the $Pt_2Ir_1$/CoP catalysts by displaying the typical CO stripping peak characteristics of PtIr alloy, rather than the individual Pt and Ir CO stripping peak characteristics (Supplementary Fig. 6)[40].

To experimentally explore the fundamental electrocatalytic behavior of the above model catalysts, we carried out the linear sweep voltammetry (LSV) via a standard three-electrode system in Ar-saturated 0.5 M $H_2SO_4$ solution. In line with the previously reported protocols, all overpotentials here were iR-corrected and calibrated to the reversible hydrogen electrode (RHE) scale (see "Methods" section for experimental details). Initially, the catalytic performance of various PtIr/CoP electrocatalysts with different molar ratios of Pt/Ir was evaluated. As shown in Supplementary Fig. 7 and Supplementary Table 1, the $Pt_3Ir_1$/CoP, $Pt_2Ir_1$/CoP, $Pt_1Ir_1$/CoP, $Pt_1Ir_2$/CoP, and $Pt_1Ir_3$/CoP electrocatalysts showed similar structural features and chemical compositions. The V-shaped relationship between the HER activity and the Pt/Ir molar ratio revealed that the best catalytic HER performance was observed on $Pt_2Ir_1$/CoP (Supplementary Fig. 8). Afterward, the loadings of $Pt_2Ir_1$/CoP were assessed. It was recognized that the chemical and morphological characters, especially the size of the loaded metal in $Pt_2Ir_1$/CoP with various metal loadings (from 0.5 to 2.0 wt%), were similar (Fig. 3a and Supplementary Fig. 9 and Supplementary Table 2). Therefore, their influences could be excluded. Revealed from the catalytic evaluation (Supplementary Fig. 10), $Pt_2Ir_1$/CoP showed evident activity improvement with the increase of metal loadings. It should be noted that such loading increment reached a threshold (herein, 1.0 wt%) and afterward delivered the limited activity improvement, suggesting an optimum metal loading at 1.0 wt%. Besides, compared with the Pt/CoP (Supplementary Fig. 11), Ir/CoP (Supplementary Fig. 11) and $Pt_2Ir_1$/CoP (Fig. 3) catalysts with the similar structural features at the same metal loading of 1.0 wt%, the $Pt_2Ir_1$/CoP catalysts delivered the greatly highest catalytic activity (Supplementary Fig. 12), demonstrating the significance of the alloyed metals for the improved catalytic activity.

To experimentally examine the above proposed fundamental understandings on the hydrogen spillover phenomenon, $Pt_2Ag_1$/

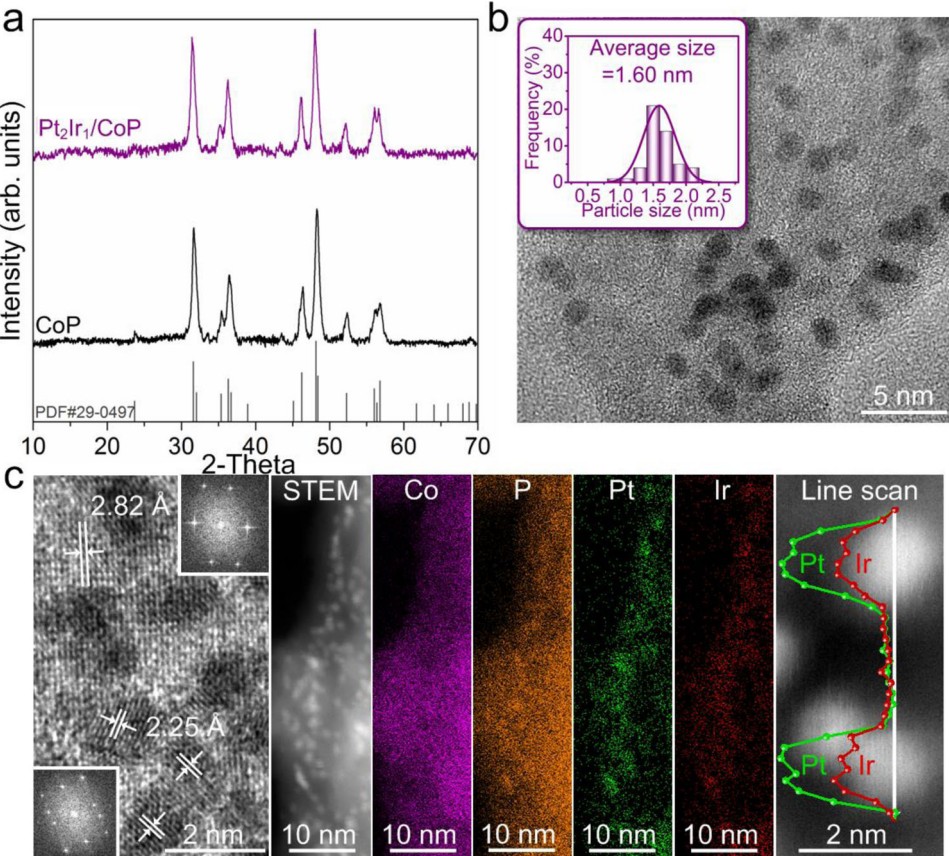

**Fig. 3 Characterizations for a paradigm of PtM/CoP catalyst. a** XRD patterns for the $Pt_2Ir_1$/CoP and CoP. **b** TEM images for the $Pt_2Ir_1$/CoP. **c** HR-TEM images, elemental X-ray mapping and line scan for the $Pt_2Ir_1$/CoP.

CoP, $Pt_2Rh_1$/CoP, $Pt_2Au_1$/CoP, Pt/CoP, $Pt_2Pd_1$/CoP, and $Pt_2Ir_1$/CoP catalysts with the same metal loading of 1.0 $wt$% were synthesized and evaluated for HER (If not otherwise specified, all $Pt_2M_1$/CoP catalysts correspond to the metal loading of 1.0 $wt$%). As shown in Supplementary Fig. 13 and Supplementary Table 3, all catalysts exhibited nearly the same structural features including metal sizes, morphologies, and phases of catalysts, enabling a straightforward comparison of their electrocatalytic performance only by considering the chemical composition of the alloyed metals.

As predicted for the most promising model catalyst, the $Pt_2Ir_1$/CoP catalysts delivered the highest catalytic activity with the lowest overpotential ($\eta_{20}$) of 7 mV at 20 mA·cm$^{-2}$ and the smallest Tafel slope of 25.2 mV·dec$^{-1}$ (Fig. 4a and b), which comprehensively surpassed the commercial benchmarks and the most of the state-of-the-art HER electrocatalysts (Supplementary Tables 4, 5). Their HER activity was also assessed in $H_2$-saturated 0.5 M $H_2SO_4$ to ensure the $H_2/H_2O$ equilibrium for HER (details can be found in "Methods" section and Supplementary Fig. 14)[41,42], which presented the superior HER activity ($\eta_{20}$ = 9 mV and Tafel slope = 25.0 mV·dec$^{-1}$) among the state-of-the-art HER electrocatalysts (Supplementary Table 6). Especially, by normalizing to the noble-metal weight loading, the $Pt_2Ir_1$/CoP catalysts gave a mass activity as high as 110 A·mg$_{PtIr}$$^{-1}$ at overpotential of −50 mV vs RHE, which was 78 times higher than that of the commercial HER catalyst (20 $wt$% Pt/C, 1.4 A·mg$_{Pt}$$^{-1}$) and even an order of magnitude higher than that for the state-of-the-art noble-metal HER electrocatalysts under the similar operation conditions (Fig. 4c)[21,23,26,43–56]. Also, the mass activity of the state-of-the-art Pt benchmark catalysts (18 A·mg$_{Pt}$$^{-1}$ at −30 mV vs RHE)[57] was still lower than that for $Pt_2Ir_1$/CoP

catalysts (78 A·mg$_{PtIr}$$^{-1}$ at −30 mV vs RHE) at similar noble metal weight loading (0.6 μg·cm$^{-2}$ for $Pt_2Ir_1$/CoP and 0.5 μg·cm$^{-2}$ for Pt benchmark). These values represent the very efficient utilization of noble metal for HER in this investigation, demonstrating the importance of our primary understandings on the hydrogen spillover of binary metal/support catalysts.

The $Pt_2Ir_1$/CoP also displayed high catalytic HER durability without an obvious decay for 50,000 cycles as well as a period of 500 h (Fig. 4d). The characterizations, including XPS, ICP-MS, and HRTEM of the spent $Pt_2Ir_1$/CoP-1.0 electrocatalysts after the durability test (Supplementary Figs. 15, 16), indicate that the structure and composition underwent negligible changes. Also, the negligible Faraday efficiency loss was observed for $Pt_2Ir_1$/CoP during HER (inset in Fig. 4d). These results demonstrate the catalytic robustness of $Pt_2Ir_1$/CoP towards HER and their potentials for practical applications.

**Correlation between ΔΦ and HER activity**. To gain further insights on the relationship between the ΔΦ and HER activity of the above catalysts, the activity parameters ($\eta_{20}$ and Tafel slopes) of the six prepared electrocatalysts were compared as a function of ΔΦ (Fig. 5a and b). The plots of the observed $\eta_{20}$ and ΔΦ values of various catalysts displayed a nearly linear decreasing trend, which could be expressed as $\eta_{20}$ = 16.2 + 470.5 ΔΦ. The derived Tafel slopes of various electrocatalysts exhibited a linear increasing trend as the function of ΔΦ, expressed as $Tafel\ slope$ = 21.3 + 324.8 ΔΦ. Both plots representatively demonstrated the highest activity of $Pt_2Ir_1$/CoP because of the fastest kinetics in the linear trend at the minimum ΔΦ value of ~0.02 eV. As a comparison, the initial Pt/CoP with median ΔΦ of ~0.19 eV corresponded to the mediocre activity and kinetics. In summary, our

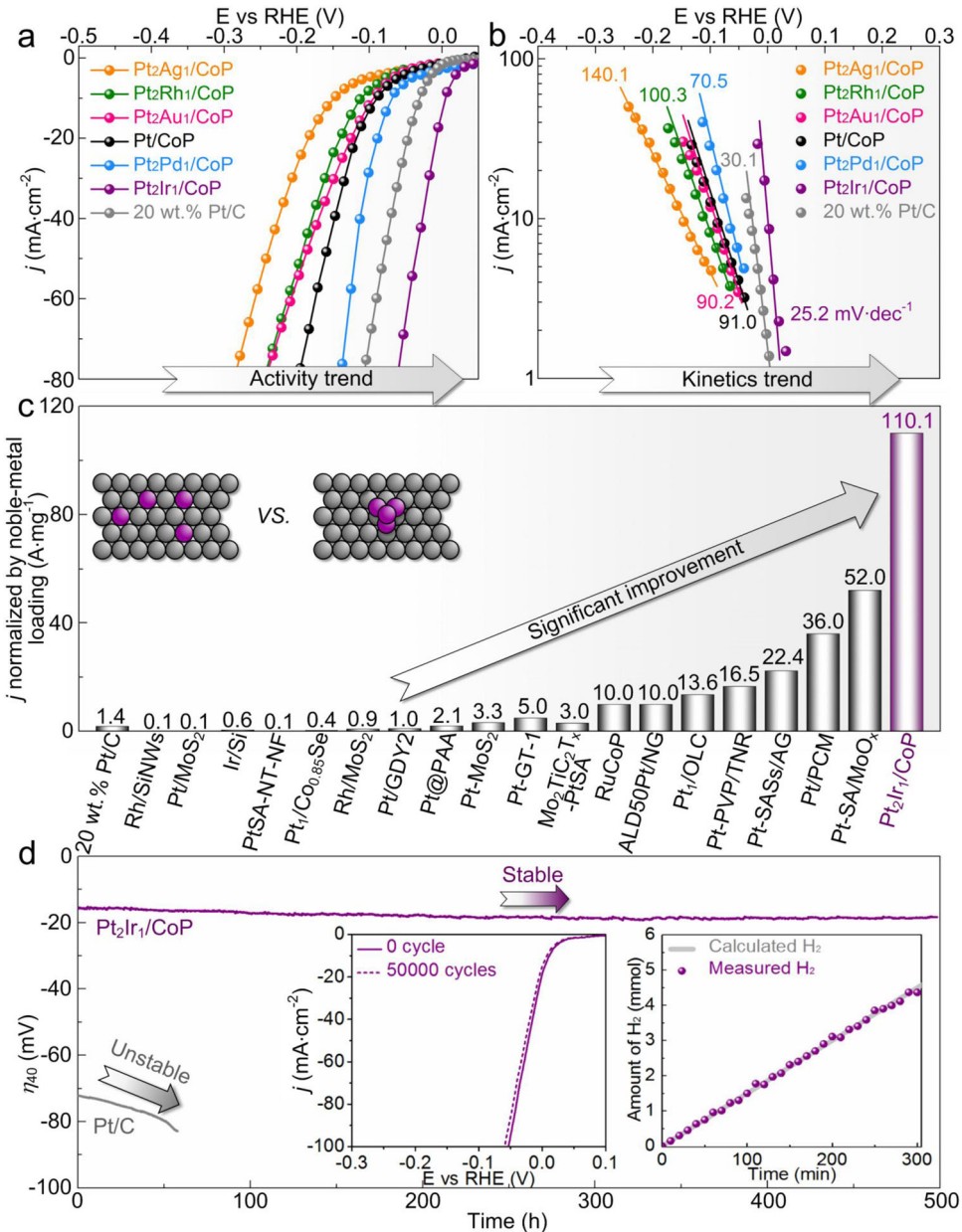

**Fig. 4 Catalytic evaluation of PtM/CoP with a total metal loading of 1.0 $wt$% in 0.5 M $H_2SO_4$. a** LSV curves of various $Pt_2M_1$/CoP and Pt/CoP catalysts (1.0 $wt$%) and Pt/C (20 $wt$.%) benchmarks. **b** LSV–derived Tafel plots for various catalysts. **c** Comparisons of the noble-metal utilization activity of $Pt_2Ir_1$/CoP (1.0 $wt$%) for HER with those of other well-known noble-metal based catalysts, especially the single-atom Pt catalysts[21,23,26,43–56]. **d** Catalytic durability of $Pt_2Ir_1$/CoP (1.0 $wt$%) through a time-overpotential profile at 40 mA·cm$^{-2}$. Insets are cycle performance of LSV curves and Faradic efficiency.

observations confirm the strong dependence of the $\Delta\Phi$ values of various PtM/CoP on their HER catalytic activity.

To further understand this relationship, the $\eta_{20}$ and Tafel slopes of Pt/CoP and $Pt_2Ir_1$/CoP catalysts with various metal loading were compared (Fig. 5c and d). By increasing Pt loading from 0.5 $wt$% to 2.0 $wt$%, the above activity parameters of Pt/CoP showed little variation compared with those of bare CoP ($\eta_{20}$ = 150 mV and Tafel slope = 108.1 mV·dec$^{-1}$). Clearly, the CoP acts as the dominant catalytical active species and the kinetic bottleneck of hydrogen adsorption remains[22,58]. Comparatively, the quickly decreased $\eta_{20}$ and Tafel slopes of $Pt_2Ir_1$/CoP with the increase of metal loadings suggest a kinetically favorable HER process, in which the hydrogen adsorption on catalyst surfaces with a low metal loading is no longer the bottleneck. Thus, such a significant HER activity improvement of $Pt_2Ir_1$/CoP

(Supplementary Fig. 12) might be explained by three possible pathways:[22,59–61] (a) $Pt_2Ir_1$ itself delivers the dominant HER contributions owing to the alloying effects; (b) CoP itself exhibits the main HER contributions owing to the electronic metal-support interaction; (c) hydrogen spillover shows the leading HER contributions owing to the integration of the beneficial proton adsorption on the alloyed PtIr and the easy hydrogen desorption on CoP.

Generally, the previously reported metal-supported HER catalysts have either high metal loadings or atomic-sized metals (Supplementary Tables 4 and 6). These characters are beneficial to achieve the abundant HER catalytic sites as well as strong electronic modulation due to the metal-support interaction, which are the prerequisites for the HER activity improvement through pathway (a) and/or (b). Considering the very low total

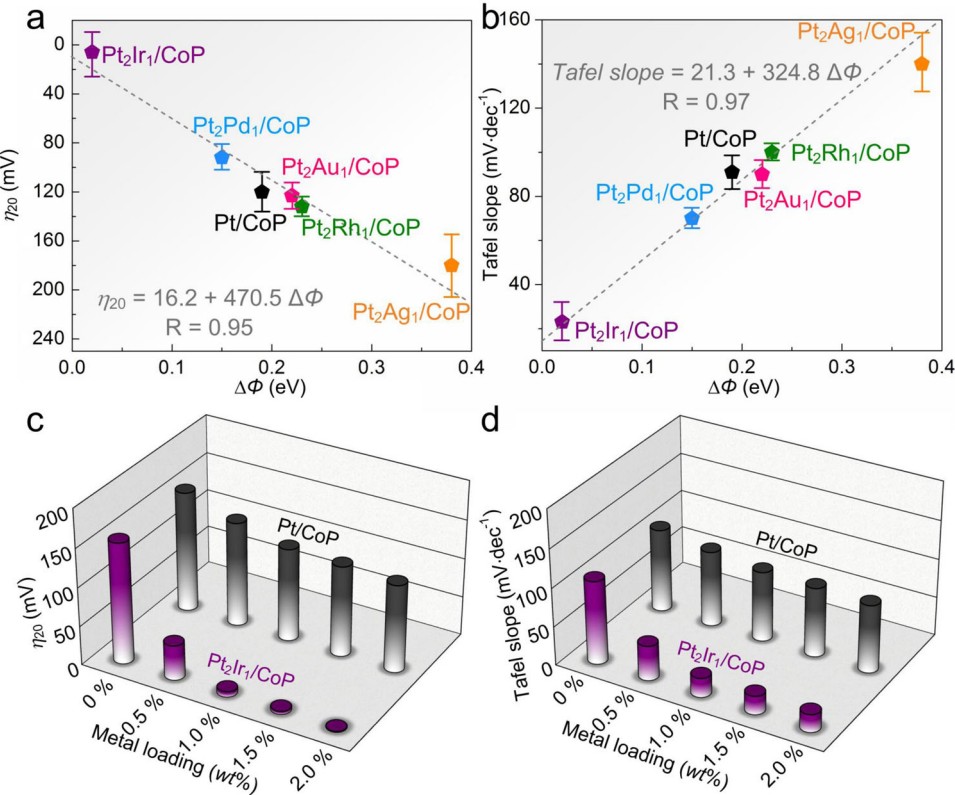

**Fig. 5 Correlation with intrinsic HER activity of various PtM/CoP model catalysts and their $\Delta\Phi$. a** HER activity trends of $\eta_{20}$ as a function of the $\Delta\Phi$.
**b** Plots of LSV–derived Tafel slope values as a function of the $\Delta\Phi$. Error bars stand for the standard deviation of three independent HER measurements.
Comparisons of $\eta_{20}$ (**c**) and Tafel slope (**d**) of $Pt_2Ir_1$/CoP paradigm with Pt/CoP as positive control catalysts at each metal loading.

$Pt_2Ir_1$ loading (the 1.0 *wt%* loading in catalyst as well as the 0.6 $\mu g \cdot cm^{-2}$ loading in the fabricated electrodes) and the large $Pt_2Ir_1$ size (~1.6 nm) of $Pt_2Ir_1$/CoP catalysts in comparison with those catalysts in previous reports, $Pt_2Ir_1$ or CoP itself in $Pt_2Ir_1$/CoP catalysts in the current work would not contribute profoundly to the overall HER activity.

To experimentally examine this point and to understand the nature behind the HER activity improvement, various control experiments have been designed and carried out. Initially, we have synthesized the control catalysts by loading Pt and $Pt_2Ir_1$ nanoparticles on reduced graphene oxide (Pt/rGO and $Pt_2Ir_1$/rGO) through the similar approach to examine the contributions of pathway (a) (details can be found in Supplementary Methods). As revealed from TEM images (Supplementary Figs. 17-18), the chemical and morphological characters of the loaded metals in Pt/rGO and $Pt_2Ir_1$/rGO especially the weight loading (~1.0 *wt%*) and size (~1.6 nm), were similar to those of the Pt/CoP and $Pt_2Ir_1$/CoP catalysts, thereby excluding their influences on the catalytic performance. The rGO was catalytically inert for HER, consistent with the previous reports (Supplementary Fig. 19a)[62]. Thus, the apparent HER activity of Pt/rGO and $Pt_2Ir_1$/rGO could effectively reveal their catalytic contributions from the loaded metals themselves. Experimentally, the $Pt_2Ir_1$/rGO catalysts showed far higher overpotential of 193 mV to reach 20 mA·cm$^{-2}$ ($\eta_{20}$) and much larger Tafel slope of 86.2 mV·dec$^{-1}$, in comparison with those of $Pt_2Ir_1$/CoP ($\eta_{20}$ = 7 mV and Tafel slope = 25.2 mV·dec$^{-1}$) and even CoP ($\eta_{20}$ = 156 mV and Tafel slope = 108.1 mV·dec$^{-1}$), suggesting the non-dominant contributions of $Pt_2Ir_1$ in $Pt_2Ir_1$/CoP. In addition, the $\eta_{20}$ and Tafel slope of $Pt_2Ir_1$/rGO were slightly less than that of Pt/rGO ($\eta_{20}$ = 166 mV and Tafel slope = 86.2 mV·dec$^{-1}$), further excluding the

enhanced and dominated HER activity improvement of $Pt_2Ir_1$/CoP due to the alloyed $Pt_2Ir_1$.

To investigate the catalytic contributions from pathway (b), we have incorporated thiocyanate ions (SCN$^-$) with catalysts, which are known to block and deactivate the Pt and Ir sites under acidic conditions[60,63]. As shown in Supplementary Fig. 19b, the CoP-like HER activity for SCN-$Pt_2Ir_1$/CoP ($\eta_{20}$ = 149 mV and Tafel slope = 107.1 mV·dec$^{-1}$) provided the convincing experimental evidence that the catalytic contribution of CoP itself was not primary. In addition, the HER activity of SCN-$Pt_2Ir_1$/CoP was very close to that of SCN-Pt/CoP ($\eta_{20}$ = 151 mV and Tafel slope = 100.1 mV·dec$^{-1}$), again excluding the dominated contribution of CoP itself in $Pt_2Ir_1$/CoP catalysts for the HER activity improvement. The above control experiments provided strong evidences to exclude the significantly enhanced HER activity of $Pt_2Ir_1$/CoP catalysts from pathway (a) and (b). Naturally, the promoted hydrogen spillover from Pt with the Ir incorporation to CoP is a most facile approach to overcome the initial kinetic bottleneck for hydrogen production[21,22].

**Evidences for hydrogen spillover in $Pt_2Ir_1$/CoP.** Inspired by the previous recognition that hydrogen spillover was highly relevant to the properties of supports in heterogeneous catalysis[20,64], it was expected that the hydrogen adsorption and desorption behavior on CoP support in $Pt_2Ir_1$/CoP catalysts should be substantially different if the hydrogen spillover phenomenon indeed exists. In situ monitoring the hydrogen adsorption and desorption behaviors on catalysts can provide strong evidence to verify the occurrence of hydrogen spillover in $Pt_2Ir_1$/CoP catalysts. To examine the hydrogen adsorption behavior, the *operando* electrochemical impedance spectra (EIS) investigations were

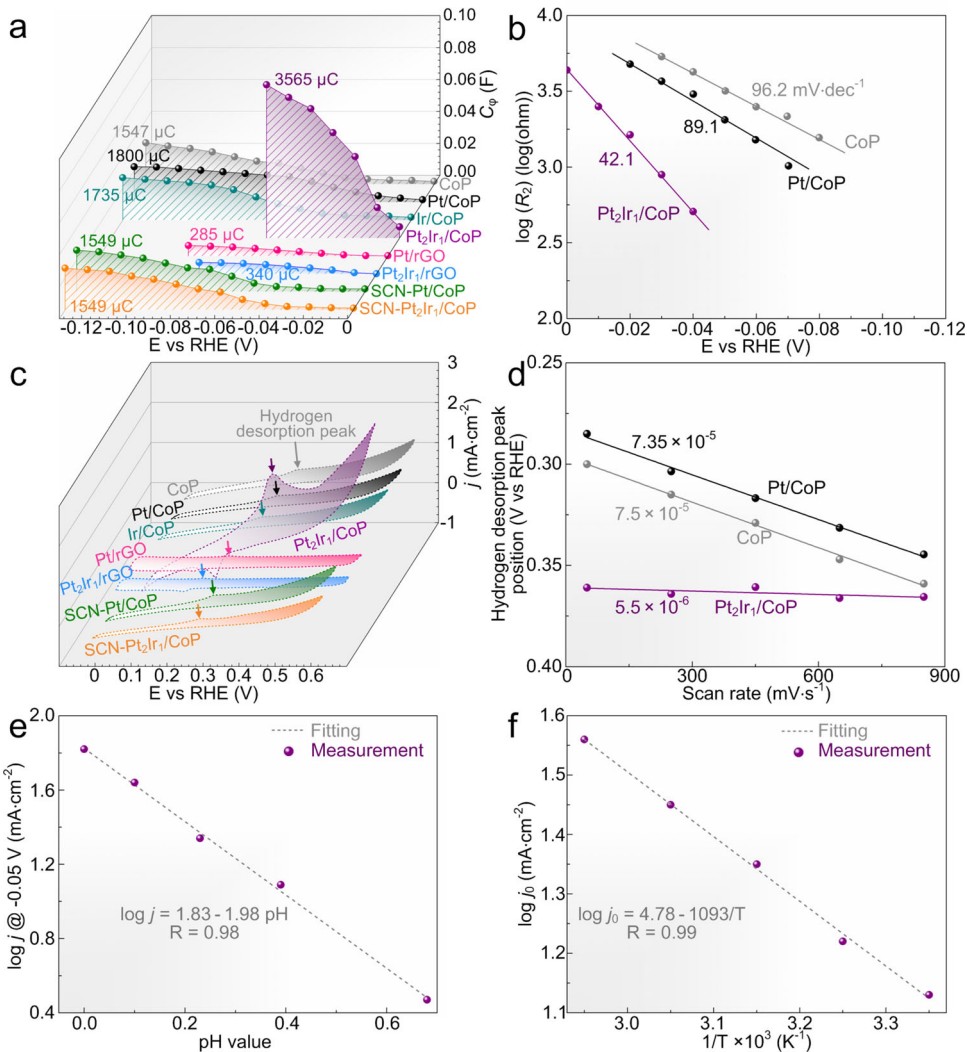

**Fig. 6 Hydrogen adsorption and desorption behaviors of various catalysts. a** Plots of $C_\varphi$ vs. $\eta$ of the bare CoP, Pt/CoP, Ir/CoP, $Pt_2Ir_1$/CoP, Pt/rGO, $Pt_2Ir_1$/rGO, SCN-Pt/CoP, and $Pt_2Ir_1$/CoP catalysts during HER in 0.5 M $H_2SO_4$. **b** EIS-derived Tafel plots of the bare CoP, Pt/CoP and $Pt_2Ir_1$/CoP catalysts obtained from the hydrogen adsorption resistance $R_2$. **c** CV of the bare CoP, Pt/CoP, Ir/CoP, $Pt_2Ir_1$/CoP, Pt/rGO, $Pt_2Ir_1$/rGO, SCN-Pt/CoP and SCN-$Pt_2Ir_1$/CoP catalysts in 0.5 M $H_2SO_4$ with a scan rate of 50 mV/s. **d** Plots of hydrogen desorption peak position vs. scan rates of the bare CoP, Pt/CoP, and $Pt_2Ir_1$/CoP catalysts. **e** Plots of log $j$ at −0.05 V (vs. RHE) vs. pH for $Pt_2Ir_1$/CoP catalysts. **f** Typical Arrhenius plots for $Pt_2Ir_1$/CoP catalysts.

carried out on bare CoP, Pt/CoP, Ir/CoP, $Pt_2Ir_1$/CoP, Pt/rGO, $Pt_2Ir_1$/rGO, SCN-Pt/CoP, and SCN-$Pt_2Ir_1$/CoP catalysts at different overpotentials (Supplementary Fig. 20). The recorded Nyquist plots were simulated by a double-parallel equivalent circuit model (insets of Supplementary Fig. 20 and Supplementary Table 7)[65–67]. The first parallel components ($T$ and $R_1$) reflect the charge-transfer kinetics, in which $T$ is related to the double layer capacitance and $R_1$ represents the catalytic charge-transfer resistance. The small values of potential-independent $R_1$ for all catalysts suggest CoP as a good conductive network in all catalysts and thereby create a fast charge-transfer kinetics for HER.

The second parallel components ($C_\varphi$ and $R_2$) reflect the hydrogen adsorption behavior on catalyst surface, where $C_\varphi$ and $R_2$ represent the hydrogen adsorption pseudo-capacitance and resistance, respectively. As shown in Fig. 6a, the integration of $C_\varphi$ vs. $\eta$ profiles provides information on the hydrogen adsorption charge ($Q_H$) on the catalyst surface during HER[22]. The almost equally small $Q_H$ values of Pt/rGO ($Q_H[Pt/rGO] = 285$ μC), $Pt_2Ir_1$/rGO ($Q_H[Pt_2Ir_1/rGO] = 340$ μC) and Ir on Ir/CoP ($Q_H[Ir/CoP]-Q_H[CoP] = 188$ μC), as well as the similar $Q_H$ values of bare

CoP ($Q_H[CoP] = 1547$ μC), SCN-Pt/CoP ($Q_H[SCN-Pt/CoP] = 1549$ μC) and SCN-$Pt_2Ir_1$/CoP ($Q_H[SCN-Pt_2Ir_1/CoP] = 1576$ μC) further support that the enhancement from pathway (a) and (b) for $Pt_2Ir_1$/CoP catalysts was too slight to dominate their hydrogen adsorption behavior and HER activity improvement. The barely increased amount of the adsorbed hydrogen on CoP of Pt/CoP ($Q_H[Pt/CoP]-Q_H[Pt/rGO] = 1515$ μC) in comparison with that on bare CoP ($Q_H[CoP] = 1547$ μC) rationalized the limited hydrogen spillover from Pt to CoP. Comparatively, the exponentially increased amount of the adsorbed hydrogen on CoP of $Pt_2Ir_1$/CoP ($Q_H[Pt_2Ir_1/CoP]-Q_H[Pt_2Ir_1/rGO] = 3225$ μC) in comparison with that of Pt/CoP ($Q_H[Pt/CoP]-Q_H[Pt/rGO] = 1515$ μC) strongly suggested the highly promoted hydrogen spillover from $Pt_2Ir_1$ to CoP as an effortless acquisition path for the hydrogen adsorption of CoP.

In this way, the corresponding hydrogen adsorption kinetics should also change. Considering the potential-dependent $R_2$ for all catalysts, it is rational to quantify their hydrogen adsorption kinetics by plotting log $R_2$ vs. overpotential and calculate the EIS-derived Tafel slopes by virtue of the Ohm's law[67]. As shown in Fig. 6b, the similar EIS-derived Tafel slope of Pt/CoP compared

with that of bare CoP suggests its unaltered hydrogen adsorption kinetics. Hence, Pt/CoP showed the individual hydrogen adsorption on respective Pt and CoP, supporting the limited hydrogen spillover from Pt to CoP due to the sluggish spillover kinetics. Comparatively, the significantly declined EIS-derived Tafel slope for $Pt_2Ir_1$/CoP indicates an accelerated hydrogen adsorption kinetics. Such a phenomenon revealed that the intrinsically insufficient hydrogen adsorption on CoP was significantly improved by a successful hydrogen spillover from $Pt_2Ir_1$ to CoP. This spillover provides a faster pathway for hydrogen adsorption owing to the profoundly enhanced spillover kinetics (better than the kinetics of solo hydrogen adsorption on CoP).

To investigate the hydrogen desorption behavior, *operando* cyclic voltammetry (CV) investigations were also carried out on bare CoP, Pt/CoP, Ir/CoP, $Pt_2Ir_1$/CoP, Pt/rGO, $Pt_2Ir_1$/rGO, SCN-Pt/CoP, and SCN-$Pt_2Ir_1$/CoP catalysts and their hydrogen desorption peak was monitored during CV scanning in the double layer region[68–71]. As shown in Fig. 6c, the CV curves showed that the intensity of the hydrogen desorption peaks for Pt/rGO and $Pt_2Ir_1$/rGO as well as bare CoP, Pt/CoP, and Ir/CoP was equally weak. Similar characteristics were also found on the hydrogen desorption peaks of bare CoP, SCN-Pt/CoP, and SCN-$Pt_2Ir_1$/CoP. The above facts further support that the enhancement from pathway (a) and (b) for $Pt_2Ir_1$/CoP catalysts was too weak to dominate their hydrogen desorption behavior. The similar hydrogen desorption peak of Pt/CoP compared with that of bare CoP suggests its almost non-increased amount of desorbed hydrogen, implying limited or no hydrogen spillover from Pt to CoP and thus the lack of abundant spillovered hydrogen for desorption. Comparatively, the significantly stronger hydrogen desorption peak of $Pt_2Ir_1$/CoP undoubtedly illustrated the highly promoted hydrogen spillover from $Pt_2Ir_1$ to CoP and thus the excessive spillovered hydrogen on CoP for efficient desorption.

The findings as discussed above could also be verified by analyzing the corresponding hydrogen desorption kinetics. The CV curves of the bare CoP, Pt/CoP, and $Pt_2Ir_1$/CoP catalysts showed a hydrogen desorption peak shift depending on the scan rates because the current takes more time to respond to the applied potential in the catalysts (Supplementary Fig. 21)[72]. Thus, it is rational to quantify their hydrogen desorption kinetics by plotting hydrogen desorption peak position vs. scan rates and comparing the fitted slopes. As shown in Fig. 6d, the similar fitted slope ($7.35 \times 10^{-5}$) of Pt/CoP compared with that of bare CoP ($7.5 \times 10^{-5}$) suggests its unaltered hydrogen desorption kinetics. Comparatively, the significantly reduced slope ($5.5 \times 10^{-6}$) of $Pt_2Ir_1$/CoP suggests its drastically accelerated hydrogen desorption kinetics. It has been reported that the hydrogen desorption kinetics for the metal-supported electrocatalysts can be effectively accelerated by hydrogen spillover[68]. Therefore, the barely altered hydrogen desorption kinetics of Pt/CoP and significantly accelerated hydrogen desorption kinetics of $Pt_2Ir_1$/CoP again demonstrated the successful hydrogen spillover from $Pt_2Ir_1$ to CoP in $Pt_2Ir_1$/CoP.

To further verify hydrogen spillover in $Pt_2Ir_1$/CoP, the Tafel slope is theoretically derived as: $\frac{(2+\alpha)F}{2.303RT} = 0.023 \text{ V} \cdot \text{dec}^{-1}$ (detailed derivation given in Supplementary Methods as well as previous reports[21,26]). The agreement between this theoretical value and experimental observation (25.2 mV·dec$^{-1}$) for $Pt_2Ir_1$/CoP also confirms this hydrogen spillover-based HER pathway. Such HER pathway could be further verified by the pH-dependent HER (Fig. 6e and Supplementary Fig. 22a). In this way, the reaction order of $Pt_2Ir_1$/CoP was experimentally determined to be 1.98, which was also in accord with the theoretical value of 2.0 (see Supplementary Methods in Equation 4). Moreover, the temperature-dependent HER performance of the $Pt_2Ir_1$/CoP

catalysts was also investigated (Fig. 6f and Supplementary Fig. 22b). A linear relationship between log $j_0$ with 1/T is exhibited in a semi-logarithmic plot and then electrochemical activation energies could be calculated according to the Arrhenius equation (log $j_0 = \log(FK_c) - \Delta G_0/2.303RT$, where $R$ is the gas constant, $\Delta G_0$ is the apparent activation energy, $F$ is the Faraday constant and $K_c$ is equilibrium constant). The calculated value of $\Delta G_0$ for $Pt_2Ir_1$/CoP is 20.9 kJ·mol$^{-1}$. Such a low activation energy is beneficial to HER process. All these derived characters of $Pt_2Ir_1$/CoP are very similar to the previously reported HSBB catalysts and also strongly support the successful hydrogen spillover process and profound contributions to the HER activity with a much low metal loading as well as large metal size in acid media[21,26].

Overall, the above experiments strongly supported the occurrence of hydrogen spillover in $Pt_2Ir_1$/CoP. The $\Delta\Phi$ is experimentally verified as the index linked to the barrier of interfacial hydrogen spillover between metals and supports in HER. The large value of $\Delta\Phi$ between metal and support suggests limited hydrogen spillover and thus sluggish HER kinetics. While, the small value of $\Delta\Phi$ implies the highly promoted hydrogen spillover and thus an energetically favorable HER kinetics in the HSBB electrocatalysts.

**Theoretical modeling of hydrogen spillover in $Pt_2Ir_1$/CoP**. To better understand the fundamental efficacy of $\Delta\Phi$, density functional theory (DFT) calculations were performed to elucidate how hydrogen spillover contributes to the overall HER activity and how $\Delta\Phi$ affects the interfacial electronic states and kinetics of the interfacial hydrogen spillover. Accordingly, the energy profiles on the Pt/CoP and $Pt_2Ir_1$/CoP catalysts were compared (Fig. 7a and b). Details of the simulation models and DFT calculations are found in "Methods" section. On the Pt/CoP surface, the H* preferentially adsorbed at Pt with $\Delta G_H$ values of −0.20 eV (site 1), −0.06 eV (site 2) and −0.36 eV (site 3), respectively, suggesting significant proton trapping at the interface of Pt and CoP (site 3). Conversely, the $\Delta G_H$ for the most stable Co site (site 4) was 0.04 eV, indicating that Co site of CoP was superior to produce a hydrogen molecule by breaking the bond of Co–H. If the H can transfer across the edge of the Pt cluster to the CoP surface, the HER will be easy to proceed for both the initial adsorption of a proton and final desorption of H₂. However, such a spillover process is hindered at the interface due to the strong hydrogen capturing at the site 3 ($\Delta G_H$ = −0.36 eV). Therefore, the overall HER process is limited by the diffusion of active hydrogen species across the interface from site 3 to site 4 with the significant thermodynamic (0.40 eV) and kinetics barrier (0.79 eV).

In $Pt_2Ir_1$/CoP, it is noted that H* adsorption on Co top (site 4′, 0.05 eV) is close to the case in Pt/CoP because these Co sites are far away from PtIr alloys. Also, the changes in the $\Delta G_H$ on site 1′ and site 2′ range from −0.20 eV to −0.39 eV and from −0.06 eV to −0.15 eV, respectively, suggesting an increased hydrogen adsorption at the metal sites. When considering the solo HER on $Pt_2Ir_1$ itself, the stronger hydrogen adsorption enables faster proton supply for the reaction. However, on the other side, this also leads to the poor hydrogen desorption and slow release of active sites, still limiting the overall HER rate[7]. Importantly, the significant changes of $\Delta G_H$ at the interface (site 3′) from −0.36 eV to −0.08 eV indicates the gradually moderate hydrogen adsorption that can induce the greatly decreased thermodynamic (0.13 eV) and kinetics barrier (0.39 eV) from site 3' to site 4'. In this regard, the hydrogen spillover across the interface is greatly facilitated on the $Pt_2Ir_1$/CoP surface, leading to a highly efficient HER activity.

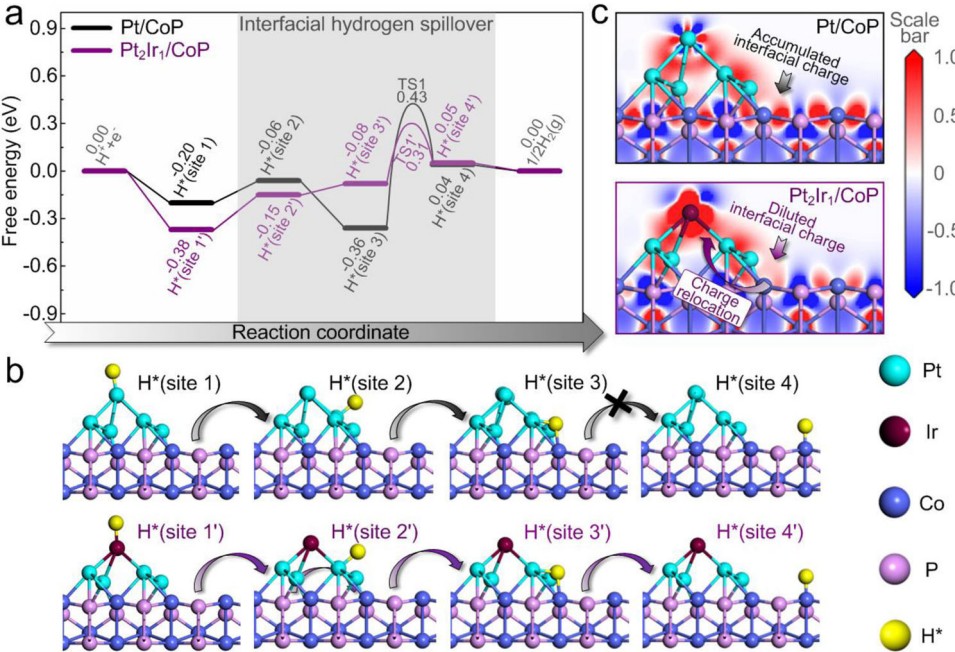

**Fig. 7 Theoretical modeling. a** Calculated free energy diagram for HER on $Pt_2Ir_1$/CoP paradigm and Pt/CoP benchmark. **b** The optimized H* adsorption structures at various sites. **c** Electron density difference map of interfaces, where a loss of electrons is indicated in blue and electron enrichment is indicated in red.

Starting from the electronic structures, we seek to further understand the relationship between the hydrogen spillover across the interface of PtIr and CoP and the $\Delta\Phi$ of binary components. The electron density difference (EDD) mapping for Pt/CoP and $Pt_2Ir_1$/CoP is shown in Fig. 7c. It is found that the high-density electron-cloud is concentrated at the interface for Pt/CoP system. Generally, active hydrogen species with the unsaturated electrons on 1s orbital is easily trapped at the electron-rich region[73]. Therefore, the electron accumulation at the interface of Pt and CoP, due to the large difference in their work functions, brings a strong bond between proton and interfacial sites, thus limiting the interfacial hydrogen spillover. Comparatively, the introduction of Ir into Pt triggers a significant relocalization of electrons, which is characterized by the vanish of the interfacially concentrated electron-cloud and directionally migrated electron-cloud towards Ir and CoP sublayer. The diluted electron-cloud at the interface of $Pt_2Ir_1$/CoP is highly consistent with the view that the small $\Delta\Phi$ can restrain the interfacial charge-transfer and minimize the interfacial electron accumulation. Such phenomena in the interfacial charge states of two catalysts are also verified by their XPS profiles (Supplementary Fig. 23). Consequently, the hydrogen adsorption atinterface site (site 3′) in $Pt_2Ir_1$/CoP was significantly weakened and gradually became thermo-neutral, which could serve as the mediators for interfacial hydrogen spillover. Furthermore, the migrated electron-cloud towards Ir will enhance the hydrogen adsorption on the Pt sites adjacent to Ir (site 1′ and site 2′) in $Pt_2Ir_1$/CoP. This situation is unfavorable for the solo HER on $Pt_2Ir_1$ itself but favorable for the interfacial hydrogen spillover because of the enhanced H* accumulation and hydrogen chemical potential on $Pt_2Ir_1$, from which the interfacial hydrogen spillover is thermodynamically facilitated[74].

Therefore, the nature of $\Delta\Phi$ between metals and supports and its contribution to HER performance are mainly embodied in two aspects (Fig. 8). (a) $\Delta\Phi$ affects the interfacial charge accumulation. Replacing Pt by PtIr alloy in Pt/CoP system will offset the intrinsic $\Delta\Phi$ between two components of the interface and thus restrain the interfacial charge flow, resulting in the reduced charge accumulation at the interface. This then enables the interfacial sites with the thermo-neutral hydrogen adsorption to become the mediators for the energetically favorable interfacial hydrogen spillover. With these functions enabled, the energy barrier for interfacial hydrogen spillover will be significantly reduced and a hydrogen spillover channel of PtIr → interface → CoP is formed. (b) $\Delta\Phi$ induces the surface charge relocation. The small $\Delta\Phi$ of binary components in PtIr/CoP results in the charge redistribution, eventually forming an electron enrichment region on metals and CoP instead of the interface. This character endows the enhanced proton adsorption on the alloyed metal sites, which leads to enhanced H* accumulation as well as hydrogen chemical potential on $Pt_2Ir_1$ and thus thermodynamically facilitates the interfacial hydrogen spillover. For these reasons, a high performance binary PtIr/CoP HER electrocatalysts are realized with a close work function, which have displayed strong hydrogen adsorption on PtIr, energetically favorable hydrogen spillover (PtIr → interface → CoP), and the efficient hydrogen desorption on CoP. All these features correspond well to our fundamental understandings on the design of HSBB catalysts. The concept of establishing advanced HER electrocatalysts through engineering the critical parameter of $\Delta\Phi$ has been substantiated through both experiments and simulations.

## Discussion
Depending on the above various control experiments and *operando* EIS and CV investigations, it is convincing that the hydrogen spillover process is the most likely reaction pathway for the significantly improved HER performance herein. However, it still lacks the strong experimental evidences by in situ tracking the spillovered hydrogen during electrocatalysis. Unfortunately, the state-of-the-art methodologies and technologies faces great challenges to in situ monitor hydrogen spillover in such metal-supported electrocatalysts due to the co-adsorption of protons on both metals and supports as well as the complicated catalytic

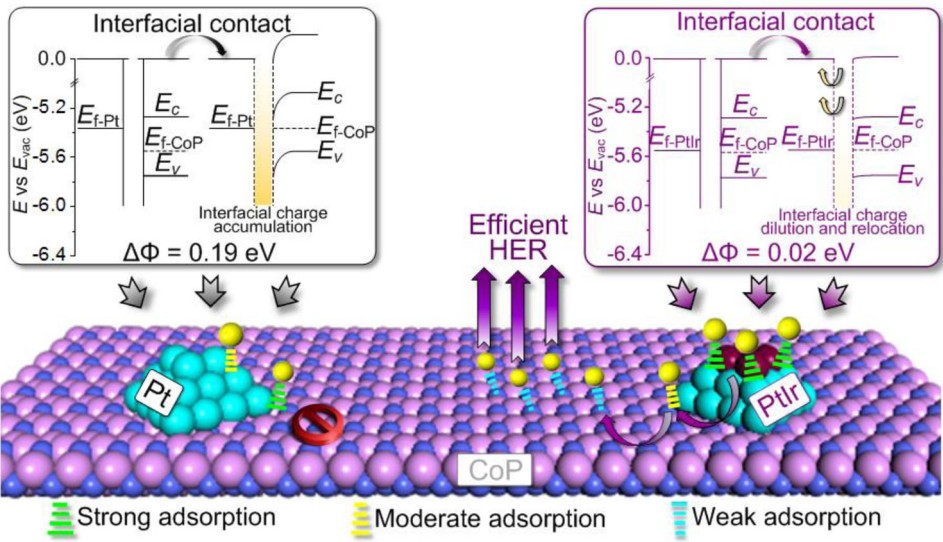

**Fig. 8 Mechanic Insights.** Proposed nature of the $\Delta\Phi$ on the hydrogen spillover phenomenon in HSBB catalysts.

environments in the presence of protons and water. The burgeoning technologies with the high spatial resolution to differentiate protons adsorbed on metals and supports and track their transfer without the interference of the electrolytes during electrocatalysis are expected to address this challenge in the future.

In this work, we theoretically and experimentally demonstrate that the work function difference between metal and support determines the interfacial electronic structures of a binary metal-supported HER electrocatalyst and thereby influences the interfacial hydrogen spillover from metal to support. Small $\Delta\Phi$ dilutes the interfacial charge density and relocates the electrons to metal and support, leading to the weakened proton adsorption at the interface and enhanced proton adsorption on metals. Thus, the significantly declined energy barrier for hydrogen transfer across the interface of metal and support enables the superior catalytic performance for HER. The hypothesis was experimentally confirmed by a series of Pt alloys-CoP hybrids with the tailorable $\Delta\Phi$, in which $Pt_2Ir_1/CoP$ (1.0 $wt$%) with the smallest $\Delta\Phi$ value exhibited the best HER performance, even better than the majority of the state-of-the-art Pt-based HER electrocatalysts as well as the commercially available Pt/C (20 $wt$.%). Our findings have not only improved the atomic understandings on the hydrogen spillover phenomenon for HER, but also pointed out a design strategy towards high performance HER electrocatalysts.

## Methods

**Synthesis of CoP.** To prepare $Co(OH)_2$ precursors, 0.582 g of $Co(NO_3)_2 \cdot 6H_2O$ and 0.56 g of hexamethylenetetramine (HMT) were dissolved in 15 mL of distilled water under the vigorous stirring to form a clear solution. The solution then was transferred into a 20 mL Teflon-lined stainless-steel autoclave and treated at 100 °C for 10 h. After cooling to room temperature naturally, $Co(OH)_2$ nanosheets were collected by centrifugation, alternatively washed by water/ethanol and dried under vacuum. To prepare CoP nanosheets, $Co(OH)_2$ nanosheets (100 mg) and $NaH_2PO_2 \cdot H_2O$ (2 g) were put at two separate positions in a quartz boat with $NaH_2PO_2$ at the upstream side of the furnace. Subsequently, the temperature of the tube furnace was raised to 300 °C with a ramping rate of 5 °C·min$^{-1}$, maintained at 300 °C for 60 min, and then naturally cooled to room temperature under the protection of Ar gas with a flow rate of 100 mL·min$^{-1}$. Then, as-synthesized CoP was collected for the future use.

**Synthesis of PtM/CoP catalysts.** Various PtM/CoP catalysts studied here were synthesized using an in situ chemical reduction of metal-salt precursors. Taking the synthesis of PtIr/CoP as an example: 50 mg of CoP was suspended in 47 ml of distilled water, and then x μL (where x = 50, 100, 150 and 200) mixture of $H_2PtCl_6$ (Pt content = 5 mg·mL$^{-1}$) and $IrCl_3$ (Ir content = 5 mg·mL$^{-1}$) solution with the desired molar ratio "y" (where y = $[Pt^{4+}]/([Ir^{3+}])$, and y = 3, 2, 1, 0.5, and 0.33)

was added to achieve the theoretical Pt: Ir molar ratio of 3: 1, 2: 1, 1: 1, 1: 2 and 1: 3 as well as the metal loadings of 0.5 $wt$%, 1.0 $wt$%, 1.5 $wt$% and 2.0 $wt$%. Subsequently, 3 mL of the freshly prepared $NaBH_4$ aqueous solution (containing 95 mg $NaBH_4$) was quickly added under vigorous stirring. After 3 h reaction under stirring, the products were collected through centrifuging, washed with water/ethanol thoroughly and then dried at 60 °C for 12 h under vacuum to obtain the PtIr/CoP catalysts. Other PtM/CoP catalysts can be obtained via the similar process when replacing the $IrCl_3$ solution by other metal-salt solution (metal content = 5 mg·mL$^{-1}$), such as $RhCl_3$, $PdCl_3$, $AgNO_3$, and $AuCl_3$. The Ir/CoP catalysts were synthesized via the similar process when only using $IrCl_3$ solution.

**Characterizations.** Ultraviolet photoemission spectroscopy was also carried out on an ESCALAB 250 Xi spectrometer using $He_I$ resonance lines (21.2 eV). ICP-MS was performed on an Agilent ICPMS 7500CE. Powder XRD data were acquired on a Shimadzu X-ray diffractometer with Cu Kα radiation. TEM and EDX measurements were performed on a JEOL 2100 F TEM with an accelerating voltage of 200 kV. XPS measurements were carried out on a Thermo Electron Model with Al Kα as the excitation source.

**DFT calculations.** In the calculations, work functions were calculated use clean and M (M = Ir, Rh, Pd, Ag, Au) doped Pt(111) surfaces. The Pt and PtM clusters were built on the CoP(011) surface with a vacuum region of 15 Å to reduce dispersive error[22]. The optimized Pt/CoP and $Pt_2Ir_1$/CoP geometries are shown in Supplementary Fig. 24. The CASTEP module of the Materials Studio software (Accelrys Inc.) was employed for the quantum chemistry calculations. Perdew-Burke-Ernzerhof Generalized-Gradient-Approximation method was used with Density Functional Dispersion correction to calculate the exchange-correlation energy[75]. Ultrasoft pseudopotential was selected to describe the characters of ionic cores. During the interface optimization, bottom CoP layers were fixed in order to reduce the calculation time. The energy cutoff is 300 eV and k-point is Gamma point for adsorption thermodynamics. The choice has been verified to produce reasonably small difference when using higher cutoff energy and larger k-point, as shown in Supplementary Fig. 25. The $4 \times 4 \times 1$ Monkhorst–Pack grids was adopted for electron density difference. LST (Linear Synchronous Transit)/QST (Quadratic Synchronous Transit) transition state search algorithm was used to search for the transition state based on nudged elastic band method. And then the transition state confirmation calculation was performed to validate the transition state[76].

The optimization is completed when the energy, maximum force, maximum stress and maximum displacement are smaller than $5.0 \times 10^{-6}$ eV·atom$^{-1}$, 0.01 eV·Å$^{-1}$, 0.02 GPa and $5.0 \times 10^{-4}$ Å, respectively. Gibbs free energy of adsorption hydrogen atom are calculated by the following equation[77].

$$\Delta G_H = E[\text{surface} + H^*] - E[\text{surface}] - 1/2E[H_2] + \Delta E_{ZPE} - T\Delta S_H$$

where $E[\text{surface}+H^*]$ is the total energy of the system, including the adsorbed molecules and facet; $E[\text{surface}]$ is the energy of the facet; $E(H_2)$ represents the total energy of a gas phase $H_2$ molecule; $\Delta E_{ZPE}$ denotes the zero-point energy of the system simplified as 0.05 eV; The $-T\Delta S_H$ is the contribution from entropy at temperature K, taken as 0.20 eV at 298 K.

**Electrochemical measurements.** Electrochemical measurements of various catalysts were conducted on a CHI 660D electrochemical analyzer (CH Instruments, Inc., Shanghai). The catalyst ink was prepared by dispersing 4 mg of various

catalysts into 1 mL of the mixed solvent containing water, ethanol, and 5% Nafion with a volumetric ratio of 768: 200:32. For the preparation of the catalytic electrodes, 5 μL of the catalyst ink was loaded onto a carbon fiber paper (CFP) electrode to reach a sample loading of ~0.06 mg·cm$^{-2}$. The CFP electrodes were dried at room temperature naturally.

HER activity of the as-prepared electrodes was investigated in a three-electrode system, using the as-fabricated electrodes as working electrode, saturated calomel electrode (SCE) as reference electrode and graphite rod as counter electrode, respectively. LSV profiles were employed to investigate the HER activities of various electrodes with a scan rate of 5 mV·s$^{-1}$ in Ar- or H$_2$-saturated 0.5 M H$_2$SO$_4$. The onset potentials were determined by the initially linear regime of the Tafel plots. The catalytic stability of the electrodes was recorded as a function of the reaction time. EIS investigations were conducted on the GAMRY Reference 600 electrochemistry workstation in the frequency range of 100 kHz-0.1 Hz at various HER overpotentials. The potentials reported herein are relative to the RHE. The calibration was performed in the high purity hydrogen saturated electrolyte with a Pt foil as the working electrode through CV investigation at a scan rate of 1 mV·s$^{-1}$[78]. The average of the two potentials at which the current crossed zero was taken to be the thermodynamic potential for the hydrogen electrode reaction. The results were shown in Supplementary Fig. 26, E(RHE) = E(SCE) + 0.281 V.

## Data availability

Data supporting the findings of this study are available within the article (and its Supplementary Information files) and from the corresponding author on reasonable request. Source data are provided with this paper.

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

## Acknowledgements

We acknowledge the financial support from China Postdoctoral Science Foundation (2018M640994 and 2018T111034) and National Nature Science Foundation of China (Grants 21872109). J. Li is supported by Postdoctoral Innovative Talents Support program (BX20180246) and Young talent Support project of Shannxi (20200601) and National Natural Science Foundation of Shaanxi province, China (No. 2019JQ486). Y. Ma is supported by National Natural Science Foundation of Shaanxi province, China (No. 2020JM-039). Jun Hu also acknowledges the financial support from the National Natural Science Foundation of China (No. 21676216), National Natural Science Foundation of Shaanxi province, China (No. 2019JM-294) and Special Project of Shaanxi Provincial Education Department (No. 20JC034).

## Author contributions

Jiayuan Li and Yongquan Qu conceived the idea, designed the experiments, conducted the characterizations, analyzed the results, and drafted the manuscript. Jun Hu contributed to the DFT calculations. Mingkai Zhang and Wangyan Gou assisted in characterizations for materials. Sai Zhang, Zhong Chen, and Yuanyuan Ma assisted in drafting the manuscript.

## Competing interests

The authors declare no competing interests.
