## [Peer Review File · Nature Communications]

REVIEWER COMMENTS

Reviewer #1 (Remarks to the Author):

In this work, the authors manufactured a series of Pt alloy nanoparticles supported on CoP supports and tried to correlate the hydrogen evolution reaction (HER) performance with the work function difference between Pt alloy nanoparticles and CoP supports. Meanwhile, the Pt₂Ir/CoP with the smallest work function difference exhibited the superior hydrogen evolution reactivity.

Nevertheless, the viewpoint of this work is unconvincing as the authors didn't provide the evidence to prove the existence of hydrogen spillover on CoP supports. Besides, the HER activity change can't be solely correlated to work function difference, considering the complicated influence of the second metal. These two aspects are vital to the conclusion of this work.

Hence, we don't think this work is suitable to be published in Nature Communications. There are several questions that the authors should solve to further improve their work so that it can be published in other journals.

1. Numerous studies manifest that the hydrogen spillover is highly relevant to the properties of supports, such as reducibility and facets (Nature 541, 68-71 (2017), Nat Nanotech 15, 848-853 (2020)). They used in-situ XAFS or control experiments to confirm the phenomenon of hydrogen spillover. In this work, how can the authors confirm the hydrogen spillover on CoP supports without any convincing experiment characterizations? We do suspect the existence of hydrogen spillover in this work. If the authors can't solve this question, the viewpoint of this work is unconvincing and may be misleading to other researchers.

2. In this work, the authors claimed that the introduction of the second metal can tune the work functions of Pt alloy nanoparticles, which then correlate the work function difference with hydrogen evolution activity. However, the authors can't rule out the activity contribution of second metal such as Ir, which is also reported as a superior HER catalyst (Adv Mater 30, 1805606 (2018)), as well as the electronic structure change of platinum alloying induced activity enhancement (J Am Chem Soc 141, 19964-19968 (2019)). The authors just discussed it in DFT calculations without any experiment results. The authors should at least design some control experiments to try to clarify it.

Reviewer #2 (Remarks to the Author):

The manuscript submitted by Ma et al. reported the synthesis and characterization of PtM/CoP catalysts (M=Ir, Rh, Pd, Ag, and Au) to fundamentally understand what are the key factors behind hydrogen spillover phenomenon for HER. The measured activity of Pt₂Ir₁/CoP catalyst is higher than even ~ other reported catalysts. Theoretical calculations are employed to obtain insight toward key factors of hydrogen spillover.

The proposed concept of work-function difference is a novel and new approach to unveil hydrogen spillover in electrochemical system, and the authors conducted characterization accordingly. Considering these point, I will recommended this paper to be published on Nat. Commun after carrying out a minor revision.

1. Measured currents during catalytic evaluation are the sum of the anodic (HOR) and cathodic (HER) currents. Because Pt-based materials are highly active catalyst for both of HER and HOR, it is important to maintain H₂ atmosphere in electrolyte during the experiment. According to

- 10.1149/2.0501514jes, conducting catalytic evaluation in Ar leads to higher onset potential due to the positive shift of the H₂/H⁺ equilibrium potential. It is necessary to provide additional HER polarization in H₂ atmosphere and compare it with other reported catalysts.
2. Some conversion issue on Fig.5 (e) (mV ?) should be revised correctly.
 3. To differentiate catalytic activity of solo Pt₂Ir₁ on HER from spillover effect, please synthesize Pt₂Ir₁ on other support materials (e.g., carbon materials) which has high conductivity but no H-uptake from Pt₂Ir₁, and measure and compare with Pt₂Ir₁ on CoP catalyst.
 4. Further details of synthesis and characterization through all stages from Co(OH)₂ nanosheets to CoP support would be beneficial to the reader. (in particular, overall morphologies, XRD, XPS, and etc.)
 5. XPS results of benchmark Pt and Ir catalysts also added into Fig. S14 and compared
 6. Line 239, the meaning of T is missing. Please explain it.
 7. As shown in Line 149-150, the XRD patterns of loaded metal could not be obtained due to the small size and low loading. Hence the physical characterization of loaded metals is somewhat insufficient, it is suggested to do further characterization (e.g., XANES, EXAFS, and etc.)

Reviewer #3 (Remarks to the Author):

The study combines in a useful way the two important phenomena, of H-spillover and of the interface charge accumulation controlled by the work function difference. Spillover used to be hot topic in the context of hydrogen storage and the authors should cite an early (prior to both [19-20]) theoretical study explaining its thermodynamics, by A.Singh et al. ACS Nano, 3, 1657, 2009. // Here, it is nice to see the simple descriptor--work function--is used again for the HER activity, I say "again" because it seems essentially similar to LUS (lowest unoccupied state) introduced as descriptor for HER by Y.Liu et al. Nature Energy, 6, 17127, DOI: 10.1038/nenergy.2017.127, 2017. Some parallels/comparison should be offered here (the authors may also read D. Chirdon and Y. Wu. Nature Energy, 6, 17132, 2017).

I think the main problem of this paper is that they do not really show there is H spillover in their system. They seem to assume that this is the case, and then study how to optimize the component materials so that the H would not be trapped at the interface. There could be other reasons (without involving the H spillover) for the observed improved performance: for example, the nanoparticle itself may get better. In fact, as stated in page 13, "the changes in the ΔG_H on site 1' and site 2' range from -0.20 eV to -0.39 eV and from -0.06 eV to -0.15 eV..." this means that by alloying the particle itself gets better. There seems to be no need to involve the interface and further H spill over. The authors must try best to clarify strengthen this part, factual evidence of logical support for spillover action, in a good revision.

Author's Response to Reviewers

Reviewer 1

In this work, the authors manufactured a series of Pt alloy nanoparticles supported on CoP supports and tried to correlate the hydrogen evolution reaction (HER) performance with the work function difference between Pt alloy nanoparticles and CoP supports. Meanwhile, the Pt₂Ir/CoP with the smallest work function difference exhibited the superior hydrogen evolution reactivity.

Nevertheless, the viewpoint of this work is unconvincing as the authors didn't provide the evidence to prove the existence of hydrogen spillover on CoP supports. Besides, the HER activity change can't be solely correlated to work function difference, considering the complicated influence of the second metal. These two aspects are vital to the conclusion of this work.

Hence, we don't think this work is suitable to be published in Nature Communications. There are several questions that the authors should solve to further improve their work so that it can be published in other journals.

Comment 1. Numerous studies manifest that the hydrogen spillover is highly relevant to the properties of supports, such as reducibility and facets (Nature 541, 68-71 (2017), Nat Nanotech 15, 848-853 (2020)). They used in-situ XAFS or control experiments to confirm the phenomenon of hydrogen spillover. In this work, how can the authors confirm the hydrogen spillover on CoP supports without any convincing experiment characterizations? We do suspect the existence of hydrogen spillover in this work. If the authors can't solve this question, the viewpoint of this work is unconvincing and may be misleading to other researchers.

Comment 2. In this work, the authors claimed that the introduction of the second metal can tune the work functions of Pt alloy nanoparticles, which then correlate the work function difference with hydrogen evolution activity. However, the authors can't rule out the activity contribution of second metal such as Ir, which is also reported as a superior HER catalyst (Adv Mater 30, 1805606 (2018)), as well as the electronic structure change of platinum alloying induced activity enhancement (J Am Chem Soc 141, 19964-19968 (2019)). The authors just discussed it in DFT calculations without any experiment results. The authors should at least design some control experiments to try to clarify it.

Response: We thank the reviewer for raising their concerns on our studies, which are

important for us to improve the quality of our studies. We would like to address his/her comments.

What we presented in this manuscript is *a fundamental understanding on hydrogen spillover phenomenon of metal-supported HER electrocatalysts and the importance of work function difference between metal and support ($\Delta\phi$) on the efficient hydrogen spillover of a Pt₂Ir₁/CoP model catalyst as well as on the prediction of the HER activities for such hydrogen-spillover-based binary (HSBB) HER electrocatalysts*. Summarized from the above comments, the reviewer raised three critical questions: *(1) Contributions of the second Ir metal; (2) Evidences of hydrogen spillover phenomenon. (3) Limited predictive power of the $\Delta\phi$ for HSBB catalysts*.

We do agree that the reviewer gave very constructive comments on our studies, which could help us to give a clear presentation and convincing evidences on these issues. Generally, a catalytic system is complicated, in which many parameters could affect the catalytic performance of a catalyst. In this revised manuscript, we would like to supply and reorganize more experimental results as well as presentation and to provide the solid experimental evidences and demonstrate the hydrogen spillover process in our study. Considering that these three questions are causally related, the response to the Comment #1 and #2 are integrated as below.

As mentioned, except for the contributions of hydrogen spillover, the improvements on HER performance for the Pt₂Ir₁/CoP catalysts may be also related to the contributions of Pt₂Ir₁ or CoP itself. This view is widely accepted while revisiting the previous efforts on the metal-supported HER electrocatalysts. For instance, Baek *et.al* (*Adv. Mater.* 2018, 30, 1805606) reported the superior HER activity of catalysts of encapsulating Ir nanoparticles inside a cage-like organic network (Ir@CON), which took benefits from the intrinsically decent HER activity of Ir species and further optimization for its spatial structures by CON. Wu *et. al* (*J. Am. Chem. Soc.* 2019, 141, 19964-19968) reported the catalysts of depositing Pt submonolayer on an intermetallic Pd₃Pb nanoplate (AL-Pt/Pd₃Pb) for efficient HER electrocatalysis, which could be ascribed to the largely optimized atomic efficiency and electronic structure of the catalytically active Pt layer. Yang *et. al* (*Energy Environ. Sci.*, 2020, 13, 3110) reported that the catalysts of introducing single-atom Pt dopant into the Co₂P catalysts (Pt-Co₂P) significantly optimized the electronic structures and thereby HER process of Co₂P, affording the superior HER performance. Clearly, the catalysts

mentioned above have either high metal loading (Ir@CON, 22 wt.%) or small metal size (Pt-Co₂P and AL-Pt/Pd₃Pb, atomic scale). These characters were beneficial to achieve abundant HER catalytic sites on metal as well as strong electronic metal-support interaction, which were the preconditions of achieving the profound contributions of metal or support itself for the overall HER activity. Considering the low total Pt₂Ir₁ loading (1.0 wt.%) and the large Pt₂Ir₁ size (~ 1.6 nm) in Pt₂Ir₁/CoP catalysts, it was expected that Pt₂Ir₁ or CoP itself in Pt₂Ir₁/CoP catalysts would not contribute profoundly to the overall HER activity. To experimentally examine the roles of Pt₂Ir₁ or CoP and figure out the nature behind the HER activity improvements, various control experiments have been designed and carried out.

(1) Contributions of the second Ir metal: the catalytic performance of various control catalysts.

(a) Initially, we have supplemented the control catalysts by loading Pt and Pt₂Ir₁ nanoparticles on reduced graphene oxide (Pt/rGO and Pt₂Ir₁/rGO) through the similar approach in the initial submission to examine the contributions of metals for such high catalytic performance of Pt₂Ir₁/CoP. It was found that the chemical and morphological characters of the loaded metals in Pt/rGO and Pt₂Ir₁/rGO, especially the loading (~ 1.0 wt.%) and size (~ 1.58 nm) were similar (Figure R1 and R2) to the catalysts of Pt/CoP and Pt₂Ir₁/CoP, therefore excluding their influences on the catalytic performance for various catalysts.

(a) High-resolution X-ray photoelectron spectra (XPS) spectrum in Pt 4f region for Pt/rGO catalysts. (b) Transmission electron microscopy (TEM) images for the Pt/rGO catalysts. (c) Dark-field scanning transmission electron microscope (STEM) images and (d) energy-dispersive

X-ray (EDX) mapping for the Pt/rGO catalysts. (e) Inductively-coupled plasma mass spectrometry (ICP-MS) analysis for the Pt/rGO catalysts.

Figure R2 | Characterizations for the Pt₂Ir₁/rGO catalysts. (a) High-resolution XPS spectra in Pt 4f and Ir 4f regions. (b) TEM images. (c) STEM images, EDX and line scan. (d) ICP-MS analysis.

Figure R3 | Catalytic contribution analysis. (a) HER performance comparison of Pt/CoP, Pt₂Ir₁/CoP, Pt/rGO and Pt₂Ir₁/rGO catalysts. (b) HER performance comparison of Pt/CoP and Pt₂Ir₁/CoP catalysts in the presence or absence of the thiocyanate ions (SCN⁻) probe.

In Figure R3a, the rGO was catalytically inert for HER, consistent with the previous reports (*J. Am. Chem. Soc.* 2011, 133, 7296–7299; *Adv. Funct. Mater.* 2016, 26, 6785–6796). Thus, the apparent HER activity of Pt/rGO and Pt₂Ir₁/rGO could effectively embody the catalytic contributions of the loaded metals themselves. Experimentally, the Pt₂Ir₁/rGO catalysts showed far higher overpotential of 193 mV to reach 20 mA/cm² (η_{20}) and much larger Tafel slope of 86.2 mV/dec, in comparison with those of Pt₂Ir₁/CoP (η_{20} = 7 mV and Tafel slope = 25.2 mV/dec) and even CoP

($\eta_{20} = 156$ mV and Tafel slope = 103 mV/dec), suggesting the non-dominant contributions of Pt₂Ir₁ in Pt₂Ir₁/CoP. In addition, the η_{20} and Tafel slope of Pt₂Ir₁/rGO were slightly less than that of Pt/rGO ($\eta_{20} = 166$ mV and Tafel slope = 86.2 mV/dec), **further excluding the case that Pt₂Ir₁ itself in Pt₂Ir₁/CoP catalysts enhanced and dominated the HER activity improvements in our catalytic system.** These results could also correspond to the DFT calculations in the revised manuscript (Figure 7a). As the reviewer noticed, compared with the Pt/CoP model, the Pt₂Ir₁/CoP showed the changes in the ΔG_H on Pt₂Ir₁ itself (site 1': -0.20 eV \rightarrow -0.39 eV; site 2': -0.06 eV \rightarrow -0.15 eV), indicating the stronger hydrogen adsorption at the Pt₂Ir₁ sites. When considering the separate HER on Pt₂Ir₁ itself, the stronger hydrogen adsorption enables faster proton supply for the reaction, however, on the other side, this leads to weaker hydrogen desorption and slower release of active sites, still limiting the overall HER rate (*Science*, 2017, 355, eaad4998). From this view, the less HER activity of Pt₂Ir₁/rGO relative to Pt/rGO is predictable.

(b) To investigate the catalytic contributions of the CoP itself, we have incorporated thiocyanate ions (SCN⁻), which are known to block and deactivate the Pt and Ir sites under acidic conditions (*Adv. Mater.* 2018, 30, 1805606; *Energy Environ. Sci.*, 2020, 13, 4921-4929). In Figure R3b, the CoP-like HER activity for SCN-Pt₂Ir₁/CoP ($\eta_{20} = 149$ mV and Tafel slope = 107.1 mV/dec) provided the convincing experimental evidences that the catalytic contribution of CoP itself was non-dominating. In addition, the HER activity of SCN-Pt₂Ir₁/CoP was very close to that of SCN-Pt/CoP ($\eta_{20} = 151$ mV and Tafel slope = 100.1 mV/dec), **excluding the dominated contribution of CoP itself in Pt₂Ir₁/CoP catalysts for the HER activity improvements.**

Above control experiments provided strong evidences to prove the significantly enhanced HER activity on Pt₂Ir₁/CoP catalysts was not resulted from Pt₂Ir₁ or CoP itself. Reviewer also mentioned Ir as good HER catalysts (*Adv. Mater.* 2018, 30, 1805606) and Pt alloying with regulated electronic structures as the promoted HER catalysts (*J. Am. Chem. Soc.* 2019, 141, 19964-19968). The mentioned Ir@CON catalysts (*Adv. Mater.* 2018, 30, 1805606) carried out a large Ir usage and a unique strategy for optimizing the spatial structures of Ir by CON, affording the abundant catalytically Ir sites for HER. In this way, the intrinsically good HER activity of Ir could be completely utilized, thus creating superior overall HER performance.

Especially, the Ir loading in this study was over 500 $\mu\text{g}/\text{cm}^2$, which was much higher than our case with a very low Ir loading of $\sim 0.2 \mu\text{g}/\text{cm}^2$ (Table R1). In addition, the above experimental observations showed that the HER activity of Pt₂Ir₁/rGO and Pt₂Ir₁/CoP catalysts were totally different even under the same Ir loading of $\sim 0.2 \mu\text{g}/\text{cm}^2$. Thus, the roles of Ir should be different, compared to this previous study.

Table R1. Comparison of the HER activities of Pt₂Ir₁/CoP, Pt₂Ir₁/rGO and Ir@CON catalysts in 0.5 M H₂SO₄.

Catalyst	Ir loading [$\mu\text{g}/\text{cm}^2$]	η_{20} [mV]	Tafel slope [mV/dec]	Reference
Pt ₂ Ir ₁ /CoP	~ 0.2	7	25.2	Our work
Pt ₂ Ir ₁ /rGO	~ 0.2	192	86.2	Our work
Ir@CON	500	16	27.0	Adv. Mater. 2018, 30, 1805606

On the other hand, the mentioned AL-Pt/Pd₃Pb catalysts (*J. Am. Chem. Soc.* 2019, 141, 19964-19968) have an atomic-scale Pt submonolayer for HER, affording the high atomic efficiency. Upon this, the contributions of electronic structure regulation for Pt could be effectively magnified, eventually resulting in superior overall HER performance. However, the Pt utilization activity in this study was 115.6 A/mg_{Pt} at -50 mV vs RHE, which was much lower than our case with a Pt utilization activity of 165.2 A/mg_{Pt} but with a much larger particle size (~ 1.6 nm) of Pt₂Ir₁ (Table R2). In addition, it was shown that the Pt utilization activity of Pt₂Ir₁/rGO and Pt₂Ir₁/CoP catalysts were totally different even under the similar size (~ 1.6 nm) of loaded Pt₂Ir₁. All results suggested the different catalytic mechanism for such high catalytic performance of Pt₂Ir₁/CoP, compared to those previous studies. **For a better understanding, the updated discussions can be found in our revised manuscript (Page 11-14 and Page 19).**

Table R2. Comparison of the Pt utilization HER activity of Pt₂Ir₁/CoP, Pt₂Ir₁/rGO and AL-Pt/Pd₃Pb catalysts in 0.5 M H₂SO₄.

Catalyst	Size of loaded metal	j normalized by Pt loading [A/mg]	Reference
Pt ₂ Ir ₁ /CoP	~ 1.6 nm	165.2	Our work
Pt ₂ Ir ₁ /rGO	~ 1.6 nm	3.7	Our work
AL-Pt/Pd ₃ Pb	Atomic size	115.6	J. Am. Chem. Soc. 2019, 141, 19964-19968

(2) Evidences of hydrogen spillover phenomenon: the *operando* electrochemical impedance spectra (EIS) and cyclic voltammetry (CV) investigations

The reviewer proposed that hydrogen spillover was highly relevant to the properties of supports in heterogeneous catalysis and the utilization of the *operando* spectroscopy (such as in-situ XAFS) on supports could confirm the hydrogen spillover phenomenon (*Nature*, 2017, 541, 68-71; *Nat. Nanotech.* 2020, 15, 848-853). Generally, supports in heterogeneous catalysts are usually unable to realize the dissociative adsorption of hydrogen to form hydrogen intermediates under the operations. Thus, monitoring the hydrogen intermediates on supports by *operando* spectroscopy can provide the convincing evidences on the occurrence of hydrogen spillover from metal to support. Unlike those cases, in our case, hydrogen could also be adsorbed on CoP and thus bring the ambiguity whether the hydrogen intermediates on CoP originate from the hydrogen adsorption on itself or hydrogen spillover. In addition, the complex electrolyte environment (containing H₂O, H⁺, H₃O⁺ and SO₄²⁻) in electrocatalytic HER goes against the utilization of common *operando* spectroscopy. Thus, the use of common *operando* spectroscopies is impossible for our study.

Fortunately, inspired by the reviewer's opinions, it was expected that the hydrogen adsorption and desorption behavior on CoP support in Pt₂Ir₁/CoP catalysts should be substantially different if hydrogen spillover indeed exists.

(a) Hydrogen adsorption behavior. The *operando* EIS investigations were carried out on bare CoP, Pt/CoP, Pt₂Ir₁/CoP, Pt/rGO, Pt₂Ir₁/rGO, SCN-Pt/CoP and SCN-Pt₂Ir₁/CoP catalysts at different overpotentials. The recorded Nyquist plots were simulated by a double-parallel equivalent circuit model (Figure R4 and Table R3). Following a previous recognition (*Energy Environ. Sci.*, 2019, 12, 2298-2304), the second parallel components (C_{ϕ} and R_2) reflect the hydrogen adsorption behavior on catalyst surfaces, where C_{ϕ} and R_2 represent the hydrogen adsorption pseudo-capacitance and resistance, respectively.

Figure R4 | EIS analysis. Nyquist plots for (a) bare CoP, (b) Pt/CoP, (c) Pt₂Ir₁/CoP, (d) Pt/rGO, (e) Pt₂Ir₁/rGO, (f) SCN-Pt/CoP and (g) SCN-Pt₂Ir₁/CoP catalysts in 0.5 M H₂SO₄ at various HER overpotentials. Zoom-in parts were correspondingly presented as inset. The scattered symbols represent the experimental results, and the solid lines are simulation fitted results. The inset also shows the equivalent circuit for the simulation. The fitted parameters are summarized in Table R3.

Table R3 | The fitted parameters of the EIS data of bare CoP, Pt/CoP, Pt₂Ir₁/CoP, Pt/rGO, Pt₂Ir₁/rGO, SCN-Pt/CoP and SCN-Pt₂Ir₁/CoP for HER.

Catalysts	η [mV]	R_s [Ω]	T [F s ⁿ⁻¹]	R_1 [Ω]	n_1	R_2 [Ω]	C_ϕ [F]
CoP	0	3.58	0.0041	21.1	0.86	9120	0.0018
	-10	3.68	0.0039	21.0	0.90	7762	0.0024
	-20	3.63	0.0043	20.9	0.87	6310	0.0026
	-30	3.61	0.0042	20.8	0.83	5370	0.0032
	-40	3.64	0.0045	20.4	0.81	4266	0.004
	-50	3.59	0.0044	19.8	0.90	3162	0.007
	-60	3.65	0.0042	17.7	0.91	2512	0.010
	-70	3.61	0.0044	16.7	0.85	2138	0.0132
	-80	3.66	0.0042	15.8	0.86	1549	0.0145
	-90	3.57	0.0041	15.0	0.88	1191	0.0172
	-100	3.61	0.0045	14.6	0.83	879	0.0203
	-110	3.63	0.0046	13.9	0.89	616	0.0224
	-120	3.60	0.0038	12.8	0.92	340	0.024
	-130	3.54	0.0045	12.0	0.91	160	0.026
-140	3.61	0.0041	11.4	0.81	—	—	
Pt/CoP	0	3.58	0.0058	24.1	0.82	8912	0.0015
	-10	3.62	0.0062	24.0	0.91	6310	0.0025
	-20	3.57	0.0061	23.7	0.87	4786	0.004
	-30	3.56	0.0058	24.0	0.86	3715	0.0063
	-40	3.55	0.006	23.1	0.9	3020	0.0093
	-50	3.51	0.0054	22.8	0.91	2042	0.013
	-60	3.56	0.0059	21.0	0.83	1514	0.016
	-70	3.50	0.0064	20.7	0.88	1000	0.017
	-80	3.51	0.0061	20.4	0.87	735	0.018
	-90	3.49	0.006	19.8	0.93	588	0.019
	-100	3.63	0.0064	19.2	0.91	410	0.02
	-110	3.62	0.0058	18.8	0.90	299	0.021
	-120	3.55	0.0051	18.3	0.87	201	0.022
	-130	3.59	0.0065	18.0	0.86	109	0.0222
-140	3.60	0.0062	17.3	0.82	—	—	
Pt/C	0	3.42	0.0025	35.2	0.90	5623	0.0001
	-10	3.51	0.0031	34.7	0.81	3630	0.0002

	-20	3.50	0.0027	34.3	0.83	2570	0.0007
	-30	3.55	0.0039	34.2	0.83	2089	0.0016
	-40	3.53	0.0033	33.8	0.85	1380	0.0027
	-50	3.51	0.0031	33.3	0.82	1023	0.0037
	-60	3.49	0.0028	32.9	0.85	676	0.0047
	-70	3.46	0.0037	32.1	0.92	389	0.0055
	-80	3.51	0.0036	31.6	0.91	239	0.0061
	-90	3.42	0.0035	31.0	0.90	125	0.0065
	-100	3.44	0.0038	30.4	0.92	–	–
SCN-Pt/CoP	0	3.61	0.0055	25.1	0.91	8912	0.0018
	-10	3.58	0.0051	24.6	0.85	7244	0.0018
	-20	3.62	0.0057	24.2	0.86	5495	0.0020
	-30	3.60	0.0049	24.1	0.86	4466	0.0028
	-40	3.65	0.0053	23.7	0.88	3467	0.0033
	-50	3.62	0.0051	23.2	0.87	2570	0.0054
	-60	3.58	0.0058	22.8	0.81	1995	0.0097
	-70	3.54	0.0057	22.2	0.89	1479	0.014
	-80	3.55	0.0056	21.7	0.88	1149	0.015
	-90	3.66	0.0054	21.2	0.92	891	0.018
	-100	3.67	0.0049	20.5	0.81	660	0.021
	-110	3.60	0.0052	20.0	0.80	457	0.023
	-120	3.59	0.0053	19.6	0.83	281	0.025
	-130	3.61	0.0058	19.2	0.86	141	0.026
	-140	3.67	0.0056	18.7	0.80	–	–
Pt₂Ir₁/CoP	0	3.52	0.0060	24.9	0.90	4365.2	0.007
	-10	3.51	0.0054	24.2	0.92	2512	0.019
	-20	3.55	0.0055	23.6	0.81	1621.8	0.051
	-30	3.56	0.0056	23.5	0.80	891.3	0.066
	-40	3.61	0.0057	23.0	0.85	512.9	0.081
	-50	3.62	0.0059	22.6	0.82	263.2	0.088
	-60	3.49	0.0062	22.0	0.81	77.6	0.096
	-70	3.50	0.0061	21.4	0.83	–	–
Pt₂Ir₁/C	0	3.51	0.0035	36.4	0.91	4677	0.0001
	-10	3.53	0.0030	35.9	0.82	3020	0.0007
	-20	3.54	0.0037	35.5	0.82	2188	0.0024
	-30	3.45	0.0029	35.4	0.82	1259	0.0037
	-40	3.46	0.0023	35.0	0.86	871	0.0048
	-50	3.55	0.0035	34.5	0.83	646	0.0058
	-60	3.57	0.0038	34.1	0.83	389	0.0065
	-70	3.53	0.0027	33.9	0.90	246	0.0069
	-80	3.50	0.0034	32.8	0.93	100	0.0072
	-90	3.47	0.0036	32.2	0.91	–	–
SCN-Pt₂Ir₁/CoP	0	3.42	0.0045	26.3	0.81	9549	0.0010
	-10	3.51	0.0041	25.8	0.95	8511	0.0012
	-20	3.52	0.0047	25.5	0.82	6918	0.0017
	-30	3.50	0.0039	25.3	0.81	5888	0.0020
	-40	3.55	0.0043	24.9	0.80	4786	0.0038
	-50	3.52	0.0055	24.4	0.82	3631	0.0064
	-60	3.49	0.0049	23.9	0.83	2951	0.011
	-70	3.44	0.0047	23.4	0.86	2290	0.013
	-80	3.45	0.0046	22.9	0.82	1659	0.016
	-90	3.56	0.0044	22.4	0.82	1258	0.019
	-100	3.57	0.0047	21.7	0.91	851	0.021
	-110	3.50	0.0051	21.2	0.90	616	0.024
	-120	3.49	0.0050	20.8	0.93	338	0.025
	-130	3.51	0.0052	20.2	0.85	169	0.026
	-140	3.57	0.0051	19.6	0.90	–	–

As shown in Figure R5, the integration of C_{ϕ} vs. η profiles gives the hydrogen adsorption charge (Q_H) on catalyst surfaces during HER. The equally small Q_H values of Pt/rGO ($Q_H[\text{Pt/rGO}] = 285 \mu\text{C}$) and Pt₂Ir₁/rGO ($Q_H[\text{Pt}_2\text{Ir}_1/\text{rGO}] = 340 \mu\text{C}$) while similar Q_H values of bare CoP ($Q_H[\text{CoP}] = 1547 \mu\text{C}$), SCN-Pt/CoP ($Q_H[\text{SCN-Pt/CoP}] = 1549 \mu\text{C}$) and SCN-Pt₂Ir₁/CoP ($Q_H[\text{SCN-Pt}_2\text{Ir}_1/\text{CoP}] = 1576 \mu\text{C}$) further supported that the enhancement of Pt₂Ir₁ or CoP itself in Pt₂Ir₁/CoP catalysts was too slight to dominate its hydrogen adsorption behavior and HER activity improvements. Naturally, the almost non-increased amount of adsorbed hydrogen on CoP in Pt/CoP ($Q_H[\text{Pt/CoP}] - Q_H[\text{Pt/rGO}] = 1515 \mu\text{C}$) compared with that on bare CoP ($Q_H[\text{CoP}] = 1547 \mu\text{C}$) rationalized the limited hydrogen spillover from Pt to CoP. Comparatively, the exponentially increased amount of adsorbed hydrogen on CoP in Pt₂Ir₁/CoP ($Q_H[\text{Pt}_2\text{Ir}_1/\text{CoP}] - Q_H[\text{Pt}_2\text{Ir}_1/\text{rGO}] = 3225 \mu\text{C}$) compared to that in Pt/CoP ($Q_H[\text{Pt/CoP}] - Q_H[\text{Pt/rGO}] = 1515 \mu\text{C}$) strongly suggested **the existence of hydrogen spillover from Pt₂Ir₁ to CoP as a new effortless acquisition path for the hydrogen adsorption of CoP.**

Figure R5 | Plots of C_{ϕ} vs. η for bare CoP, Pt/CoP, Pt₂Ir₁/CoP, Pt/rGO, Pt₂Ir₁/rGO, SCN-Pt/CoP and Pt₂Ir₁/CoP during HER in Ar-saturated 0.5 M H₂SO₄.

In this way, the corresponding hydrogen adsorption kinetics should also change. Considering the potential-dependent R_2 for all catalysts, it is rational to quantify their hydrogen adsorption kinetics *via* plotting $\log R_2$ vs. overpotential and calculating the EIS-derived Tafel slopes by virtue of the Ohm's law (*J. Power Sources* 2006, 158, 464-476). As shown in Figure R6, the similar EIS-derived Tafel slope of Pt/CoP compared with that of bare CoP suggests its unaltered hydrogen adsorption kinetics. Hence, Pt/CoP showed the individual hydrogen adsorption on respective Pt and CoP, supporting the limited hydrogen spillover from Pt to CoP due to the sluggish spillover

kinetics. Comparatively, the significantly declined EIS-derived Tafel slope for Pt₂Ir₁/CoP indicates an accelerated hydrogen adsorption kinetics. **Such phenomenon revealed that the intrinsically insufficient hydrogen adsorption on CoP was replaced by a successful hydrogen spillover from Pt₂Ir₁ to CoP as a new faster pathway for hydrogen adsorption owing to the profoundly enhanced spillover kinetics (better than the kinetics of solo hydrogen adsorption on CoP).**

Figure R6 | EIS-derived Tafel plots for bare CoP, Pt/CoP and Pt₂Ir₁/CoP catalysts obtained from the hydrogen adsorption resistance R_2 . The solid lines represent linear fitting.

(b) Hydrogen desorption behavior. To figure out the hydrogen desorption behavior, the *operando* CV investigations were performed on bare CoP, Pt/CoP, Pt₂Ir₁/CoP, Pt/rGO, Pt₂Ir₁/rGO, SCN-Pt/CoP and SCN-Pt₂Ir₁/CoP catalysts and monitored their hydrogen desorption peak during CV scanning in the double layer region (*Angew. Chem. Int. Edit.* 2019, 58, 16038-16042; *J. Am. Chem. Soc.* 2009, 131, 14756-14760; *J. Mater. Chem. A* 2014, 2, 3954-3960; *Int. J. Hydrogen Energy* 2018, 43, 1251-1260). As shown in Figure R7, the CV curves shows that the intensities of the hydrogen desorption peaks for Pt/rGO and Pt₂Ir₁/rGO are equally weak. Similar characteristics were also found on the hydrogen desorption peaks of bare CoP, SCN-Pt/CoP and SCN-Pt₂Ir₁/CoP. Above facts further supported that the enhancement of Pt₂Ir₁ or CoP itself in Pt₂Ir₁/CoP catalysts was too slight to dominate its hydrogen desorption behavior and thereby HER activity improvements. Naturally, the similar hydrogen desorption peak of Pt/CoP compared with that of bare CoP suggests its almost non-increased amount of desorbed hydrogen, corresponding to the limited hydrogen spillover from Pt to CoP and thus the lack of abundant spillovered hydrogen for desorption. **In contrast, the significantly stronger hydrogen desorption peak of Pt₂Ir₁/CoP compared to that of Pt/CoP strongly indicated the existence of**

hydrogen spillover from Pt₂Ir₁ to CoP and thus the excess spillovered hydrogen on CoP for efficient desorption.

Figure R7 | CV curves of bare CoP, Pt/CoP, Pt₂Ir₁/CoP, Pt/rGO, Pt₂Ir₁/rGO, SCN-Pt/CoP and SCN-Pt₂Ir₁/CoP catalysts in Ar-saturated 0.5 M H₂SO₄ with a scan rate of 50 mV/s.

Above facts could also be supported by analyzing the corresponding hydrogen desorption kinetics. The CV curves of bare CoP, Pt/CoP and Pt₂Ir₁/CoP catalysts show the hydrogen desorption peak shift depending on the scan rate because the current in the catalysts takes more time to respond to the applied potential (*J. Phys. Chem. C* 2011, 115, 11880-11886). Thus, it is rational to quantify their hydrogen desorption kinetics *via* plotting hydrogen desorption peak position *vs.* scan rate and comparing the fitted slopes (Figure R8a-8c). As shown in Figure R8d, the similar fitted slope of Pt/CoP compared with that of bare CoP suggests its unaltered hydrogen desorption kinetics. Comparatively, the significantly reduced slope for Pt₂Ir₁/CoP suggests its drastically accelerated hydrogen desorption kinetics. It was reported that the hydrogen desorption kinetics for metal-support electrocatalysts could be effectively accelerated by hydrogen spillover effect (*Angew. Chem. Int. Edit.* 2019, 58, 16038-16042). **Therefore, the unaltered kinetics of the hydrogen desorption for Pt/CoP should be related to the limited hydrogen spillover from Pt to CoP, while the faster kinetics of the hydrogen desorption for Pt₂Ir₁/CoP should be originated from the efficient hydrogen spillover from Pt₂Ir₁ to CoP.**

Figure R8 | Capacitance vs. voltage profiles obtained from cyclic voltammograms of (a) bare CoP, (b) Pt/CoP and (c) Pt₂Ir₁/CoP catalysts with the scan rate from 50 to 850 mV/s in Ar-saturated 0.5 M H₂SO₄. (d) Plots of hydrogen desorption peak position vs. scan rate for bare CoP, Pt/CoP and Pt₂Ir₁/CoP catalysts. The solid lines represent linear fitting.

(c) **Consistence of theoretically calculated and experimentally measured Tafel slopes.** Referring to the previous reports (*Nat. Commun.* 2016, 7, 12272; *Adv. Funct. Mater.* 2017, 27, 1700359), the HER pathway of Pt₂Ir₁/CoP was described by the following equations:

The reaction velocity of hydrogen evolution could be written as $r = k_3\theta_{\text{CoP}-\text{H}^*}C_{\text{H}^+}$, where r is the reaction rate; k is the rate constant; θ is the hydrogen coverage of on active sites; and C_{H^+} is the concentration of hydrogen ion.

In the steady state,

$$\frac{d\theta_{\text{CoP}-\text{H}^*}}{dt} = k_2\theta_{\text{Pt}_2\text{Ir}_1-\text{H}^*}(1-\theta_{\text{CoP}-\text{H}^*}) - k_{-2}\theta_{\text{CoP}-\text{H}^*}(1-\theta_{\text{Pt}_2\text{Ir}_1-\text{H}^*}) - k_3\theta_{\text{CoP}-\text{H}^*}C_{\text{H}^+}$$

$$\frac{d\theta_{\text{Pt}_2\text{Ir}_1-\text{H}^*}}{dt} = k_1(1-\theta_{\text{Pt}_2\text{Ir}_1-\text{H}^*})C_{\text{H}^+} - k_{-1}\theta_{\text{Pt}_2\text{Ir}_1-\text{H}^*} - k_{-2}\theta_{\text{Pt}_2\text{Ir}_1-\text{H}^*}(1-\theta_{\text{CoP}-\text{H}^*}) + k_2\theta_{\text{CoP}-\text{H}^*}(1-\theta_{\text{Pt}_2\text{Ir}_1-\text{H}^*})$$

At the low overpotential,

$$\theta_{CoP-H^*} \approx \frac{k_2 \theta_{Pt_2Ir_1-H^*}}{k_2 \theta_{Pt_2Ir_1-H^*} + k_{-2} - k_{-2} \theta_{Pt_2Ir_1-H^*} + k_3 C_{H^+}} \approx \frac{k_2}{k_{-2}} \theta_{Pt_2Ir_1-H^*} e^{-\frac{F\Delta\phi}{RT}}$$

$$\theta_{Pt_2Ir_1-H^*} \approx \frac{k_1 C_{H^+} + k_{-2} \theta_{CoP-H^*}}{k_1 C_{H^+} + k_{-1} + k_2 + k_{-2} \theta_{CoP-H^*} - k_2 \theta_{CoP-H^*}} \approx \frac{k_1}{k_{-1}} C_{H^+} e^{-\frac{F\Delta\phi}{RT}}$$

Thus,

$$r \approx k_3 \theta_{CoP-H^*} C_{H^+} = \frac{k_1 k_2 k_3}{k_{-1} k_{-2}} C_{H^+}^2 e^{-\frac{(2+\alpha)F\Delta\phi}{RT}} \quad (4)$$

And,

$$-j = Fr = \frac{k_1 k_2 k_3}{k_{-1} k_{-2}} F C_{H^+}^2 e^{-\frac{(2+\alpha)F\Delta\phi}{RT}}$$

$$\lg(-j) = Constant + 2\lg C_{H^+} - \frac{(2+\alpha)F}{2.303RT} \Delta\phi$$

Therefore, the calculated Tafel slope for Pt₂Ir₁/CoP catalysts is: $\frac{(2+\alpha)F}{2.303RT} = 0.023$

V/dec (assuming $\alpha = 0.5$, F is the Faraday constant, R the Rydberg gas constant and T the absolute temperature). The coincidence between this theoretical value and experimental observation (Figure R3a, 25.2 mV/dec) for Pt₂Ir₁/CoP clearly confirmed this hydrogen-spillover-based HER pathway.

(d) pH- and temperature-dependent HER performance. Such HER pathway could be further verified by investigating the pH-dependent relation of HER (Figure R9a and R9b). In this way, the reaction order of Pt₂Ir₁/CoP was experimentally determined to be 1.98, which was also in accord with the theoretical value of 2 (Equation 4). Moreover, the temperature-dependent relation of HER for Pt₂Ir₁/CoP catalysts was also investigated (Figure R9c and R9d). A linear relationship between $\log j_0$ with $1/T$ is exhibited in a semi-logarithmic plot and then electrochemical activation energies could be calculated according to the Arrhenius equation ($\log j_0 = \log(FK_c) - \Delta G_0/2.303RT$, where R is the gas constant, ΔG_0 is the apparent activation energy, F is the Faraday constant). The calculated value of ΔG_0 for Pt₂Ir₁/CoP is 20.9 kJ/mol. Such low activation energy is beneficial to HER process. All these characters of Pt₂Ir₁/CoP corresponded well to those of the previously reported hydrogen-spillover-based HER electrocatalysts in acid media (*Nat. Commun.* 2016, 7, 12272; *Adv. Funct. Mater.* 2017, 27, 1700359) and strongly supported its successful hydrogen spillover process and profound contributions to the HER activity.

Figure R9 | (a) Tafel curves of Pt₂Ir₁/CoP catalysts in Ar-saturated H₂SO₄ with pH ranging from 0 to 0.68. (b) plots of log *j* at -0.05 V (vs. RHE) vs. pH for Pt₂Ir₁/CoP catalysts. (c) Tafel curves of Pt₂Ir₁/CoP catalysts in Ar-saturated 0.5 M H₂SO₄ at different temperatures ranging from 298 to 338 K. (d) Typical Arrhenius plots for Pt₂Ir₁/CoP catalysts. The solid lines represent linear fitting.

Depending on the well-organized evidences, **(1) the case that the enhanced Pt₂Ir₁ or CoP itself in Pt₂Ir₁/CoP catalysts dominated the HER activity improvements was excluded** and **(2) the contributions of hydrogen spillover was confirmed**. Hence, **it can be convincing that small $\Delta\phi$ for Pt₂Ir₁/CoP model catalysts in our case results in their efficient hydrogen spillover phenomenon along with superior HER activity and the $\Delta\phi$ has the predictive power for the HER activities of the PtM/CoP electrocatalysts**. For a better understanding, we have updated the relative discussion in Page 14-18.

(3) Embodiment of the predictive power of the $\Delta\phi$

To better show the predictive ability of the $\Delta\phi$ for hydrogen spillover in our studies, we also attempted to verify their efficacy with a previously reported work on other HSBB electrocatalysts. Shao *et. al* (*Adv. Funct. Mater.*, 2017, 27, 1700359) reported a series of HSBB catalysts of loading various metal (Ag, Au, Re, Ru, Rh and Pd) on molybdenum disulfide (MoS₂) support for HER, in which Rh/MoS₂ delivered the best HER activity. In this study, authors did not give a deep understanding on the best performance of Rh among all metals. Thus, we analysed their HER activity

parameters (η_{15}) as a function of $\Delta\phi$. As shown in Figure R10a, the work function of MoS₂ was determined based on the catalyst characterizations in this work as well as the previous experimental values (*J. Korean Phys. Soc. 2014, 64,1550-1555*) and the work function of loaded metal was determined by DFT calculations via the similar approach in our study. The plots of the η_{15} and $\Delta\Phi$ values of various catalysts displayed a well-fitted (R-squared = 0.94) linear decreasing trend (Figure R10b), which could confirm the strong dependence of the $\Delta\phi$ values of the metal/MoS₂ catalysts on their HER activity. These results effectively embody the predictive power of $\Delta\Phi$ for the HER activities of other HSBB catalysts.

Figure R10 | (a) The calculated work function (Φ), work function difference between metal and support ($\Delta\Phi$) and extracted overpotential (η_{15}) at 15 mA/cm² for various metal/MoS₂ catalysts. (b) HER activity trends of η_{15} as a function of the $\Delta\Phi$. The dash line represents linear fitting.

The predictive power of the proposed $\Delta\phi$ is also suitable for the previous report, which is reflected by its universal applicability on either the PtM/CoP catalysts or even other HSBB catalysts. Of course, there are many factors affecting the catalytic activity of a catalyst for a specific catalytic reaction. Considering this, the previously successes in building appropriate descriptor (*Nat. Energy 2017, 2, 17131; Nat. Commun. 2019, 10, 3755; Nat. Energy 2017, 2, 17127*) was facilitated by the idea that to separate a single variable from the various parameters and determine the barrier of this reaction by the single variable afterwards. In our work, it was noticed that the barrier of hydrogen spillover phenomenon was determined by the $\Delta\phi$ and the support, Pt and reaction conditions are identical except the alloyed second metal. It's significative to investigate the $\Delta\phi$ vs. catalytic activity. **In a word, the catalytic**

system provided a golden opportunity to illustrate the importance of *the $\Delta\Phi$ as the criterion in guiding the design of HSBB electrocatalysts for HER.* In the future, we expect to verify the reliability of the $\Delta\Phi$ in more HSBB electrocatalysts and screen the more suitable metal and support for achieving HER activity breakthroughs.

Reviewer 2

The manuscript submitted by Ma et al. reported the synthesis and characterization of PtM/CoP catalysts (M = Ir, Rh, Pd, Ag, and Au) to fundamentally understand what are the key factors behind hydrogen spillover phenomenon for HER. The measured activity of Pt₂Ir₁/CoP catalyst is higher than even other reported catalysts. Theoretical calculations are employed to obtain insight toward key factors of hydrogen spillover.

The proposed concept of work-function difference is a novel and new approach to unveil hydrogen spillover in electrochemical system, and the authors conducted characterization accordingly. Considering these points, I will recommend this paper to be published on Nat. Commun after carrying out a minor revision.

We thank the reviewer's constructive comments on our manuscript. We would like to address those comments as below.

Comment 1. Measured currents during catalytic evaluation are the sum of the anodic (HOR) and cathodic (HER) currents. Because Pt-based materials are highly active catalyst for both of HER and HOR, it is important to maintain H₂ atmosphere in electrolyte during the experiment. According to 10.1149/2.0501514jes, conducting catalytic evaluation in Ar leads to higher onset potential due to the positive shift of the H₂/H⁺ equilibrium potential. It is necessary to provide additional HER polarization in H₂ atmosphere and compare it with other reported catalysts.

Response: Thanks for the constructive suggestions. We have supplemented the linear sweep voltammetry curve for Pt₂Ir₁/CoP catalysts in H₂-saturated 0.5 M H₂SO₄ electrolyte to ensure the H₂/H₂O equilibrium for HER (*J. Electrochem. Soc.*, 2015, 162, F1470; *Adv. Mater.* 2019, 31, 1900813). As shown in Figure R1, Pt₂Ir₁/CoP catalysts deliver an overpotential of 9 mV to reach 20 mA/cm² and a Tafel slope of 25.0 mV/dec. The detailed comparison of HER activity in H₂-saturated 0.5 M H₂SO₄ electrolyte for Pt₂Ir₁/CoP with other reported catalysts is shown in Table R1. Clearly, Pt₂Ir₁/CoP presents the superior HER activity among the state-of-the-art HER electrocatalysts, especially Pt-based catalysts. **We have added the relevant discussions in the revised manuscript (Page 10).**

Figure R1 | HER activity of Pt₂Ir₁/CoP catalysts in H₂-saturated 0.5 M H₂SO₄ electrolyte.

Table R1 | Comparison of HER activity in H₂-saturated 0.5 M H₂SO₄ electrolyte for Pt₂Ir₁/CoP with the state-of-the-art HER electrocatalysts, especially Pt-based catalysts.

Catalyst	Noble metal content [$\mu\text{g}/\text{cm}^2$]	η_{20} [mV]	Tafel slope [mV/dec]	Ref.
Pt₂Ir₁/CoP	0.6	9	25.0	This work
Ni ₂ P NPs	0	130	46	J. Am. Chem. Soc. 2013 , 135 , 9267
Ni _{0.33} Co _{0.67} S ₂ NWs	0	88	44.1	Adv. Energy Mater. 2015 , 5 , 1402031
CoPS NPs	0	65	56	Nat. Mater. 2015 , 14 , 1245
B-doped CoP	0	58	50	Angew. Chem. Int. Ed. 2020 , 59 , 4154
CoP/Co-MoF	0	33	43	J. Am. Chem. Soc. 2018 , 140 , 5118
CoP ₂	0	53	32	Adv. Mater. 2019 , 31 , 1900813
AL-Pt/Pd ₃ Pb	1.6	17	18	J. Am. Chem. Soc. 2019 , 141 , 19964-19968
Pt SA/m-WO _{3-x}	0.86	76	45	Angew. Chem. Int. Ed. 2019 , 58 , 16038
Pt ₁ /OLC	1.37	55	36	Nat. Energy 2019 , 4 , 512
A-CoPt-NC	0.419	32	31	Angew. Chem. Int. Ed. 2019 , 58 , 9404
K ₂ PtCl ₄ @NC-M	5.6	15	21	Adv. Funct. Mater. 2020 , 30 , 2000531
Pt ₁ /MC	0.26	32	26	Nat. Commun. 2017 , 8 , 1490
Pt-PVP/TNR	21.9	27	27	Angew. Chem. Int. Ed. 2020 , 59 , 15902
Mo ₂ TiC ₂ T _x -PtSA	12	43	30	Nat. Catal. 2018 , 1 , 985
Pt-MoS ₂	7.0	47	25.0	Nat. Commun. 2017 , 8 , 14548
Pt-WO ₃	—	52	32.9	Nano Energy 2020 , 71 , 104653
RuCoP	60	25	31	Energy Environ. Sci. 2018 , 11 , 1819
RuP ₂ @NPC	233	59	38	Angew. Chem. Int. Ed. 2017 , 56 , 11559

Comment 2. Some conversion issue on Fig.5 (e) (mV ?) should be revised correctly.

Response: Thanks for pointing this out. We have already revised this error as shown in Figure R2. **The updated results can be found in our revised manuscript (Figure 6b).**

Figure R2 | EIS-derived Tafel plots for bare CoP, Pt/CoP and Pt₂Ir₁/CoP catalysts obtained from the hydrogen adsorption resistance R_2 . The solid lines represent linear fitting.

Comment 3. To differentiate catalytic activity of solo Pt₂Ir₁ on HER from spillover effect, please synthesize Pt₂Ir₁ on other support materials (e.g., carbon materials) which has high conductivity but no H-uptake from Pt₂Ir₁, and measure and compare with Pt₂Ir₁ on CoP catalyst.

Response: Thanks for the constructive suggestions. We have supplemented the control catalysts of loading Pt and Pt₂Ir₁ nanoparticles on reduced graphene oxide (Pt/rGO and Pt₂Ir₁/rGO) through the similar synthetic strategy in the initial submission except for replacing CoP to rGO as support. It was recognized that the chemical and morphological characters, especially the loading (~ 1.0 wt.%) and size (~ 1.58 nm) of the loaded metal in Pt/rGO and Pt₂Ir₁/rGO were similar (Figure R3 and R4) to the catalysts in the initial submission, therefore their influences can be excluded.

In Figure R3a, the rGO is catalytically inert for HER as previously reported. Thus, the apparent HER activity of Pt/rGO and Pt₂Ir₁/rGO could effectively embody the catalytic contributions of the loaded metal itself. Clearly, Pt₂Ir₁/rGO showed far large overpotential of 193 mV to reach 20 mA/cm² (η_{20}) and Tafel slope of 86.2 mV/dec relative to Pt₂Ir₁/CoP (η_{20} = 7 mV and Tafel slope = 25.2 mV/dec) and even CoP (η_{20} = 156 mV and Tafel slope = 103 mV/dec), suggesting the non-dominating contributions of Pt₂Ir₁ itself. In addition, the η_{20} and Tafel slope of Pt₂Ir₁/rGO was

even less than that of Pt/rGO ($\eta_{20} = 166$ mV and Tafel slope = 86.2 mV/dec), further excluding the case that Pt₂Ir₁ itself in Pt₂Ir₁/CoP catalysts enhanced and dominated the HER activity improvements. Thus, it was believed that hydrogen spillover effects dominated the superior HER activity of Pt₂Ir₁/CoP catalysts. **We have added the relevant discussions in the revised manuscript (Page 11-14).**

Figure R3 | Characterizations for the Pt/rGO control catalysts. (a) High-resolution X-ray photoelectron spectra (XPS) spectrum in Pt 4f region for Pt/rGO catalysts. (b) Transmission electron microscopy (TEM) images for the Pt/rGO catalysts. (c) Dark-field scanning transmission electron microscope (STEM) images and (d) energy-dispersive X-ray (EDX) mapping for the Pt/rGO catalysts. (e) Inductively-coupled plasma mass spectrometry (ICP-MS) analysis for the Pt/rGO catalysts.

Figure R4 | Characterizations for the Pt₂Ir₁/rGO control catalysts. (a) High-resolution XPS spectra in Pt 4f and Ir 4f region for Pt₂Ir₁/rGO catalysts. (b) TEM images for the Pt₂Ir₁/rGO catalysts. (c) STEM images, EDX mapping and line scan for the Pt₂Ir₁/rGO catalysts. (d) ICP-MS analysis for the Pt₂Ir₁/rGO catalysts.

Figure R5 | HER activity comparison of Pt/CoP and Pt₂Ir₁/CoP catalysts with Pt/rGO and Pt₂Ir₁/rGO control catalysts.

Comment 4. Further details of synthesis and characterization through all stages from Co(OH)_2 nanosheets to CoP support would be beneficial to the reader. (in particular, overall morphologies, XRD, XPS, and etc.)

Response: Thanks for the suggestions. Herein, the details of synthesis and characterizations from Co(OH)_2 precursors to CoP support have been supplemented. To prepare Co(OH)_2 precursors, 0.582 g of $\text{Co(NO}_3)_2 \cdot 6\text{H}_2\text{O}$ and 0.56 g of hexamethylenetetramine were dissolved in 15 mL of distilled water under the vigorous stirring to form a clear solution. The solution then was transferred into a 20 mL Teflon-lined stainless-steel autoclave. The autoclave was placed into an electric oven at 100 °C for 10 h. After cooling to room temperature naturally, Co(OH)_2 precursors were collected by centrifugation, and washed with water/ethanol alternatively and dried under vacuum. Powder XRD pattern of the product is shown in Figure R6a and consistent with that of Co(OH)_2 standard (JCPDS #02-0925), suggesting the successful synthesis of Co(OH)_2 precursors. In the high-resolution XPS of the product (Figure R6b), the typical signals of Co and O species in Co(OH)_2 were identified, further confirming the formation of Co(OH)_2 precursors (*ACS Catal.* 2015, 5, 941–947). TEM image (Figure R6c) clearly demonstrates the morphology of nanosheet for the Co(OH)_2 precursors. Overall, the Co(OH)_2 nanosheets as precursor were successfully synthesized.

Figure R6 | Characterizations for Co(OH)_2 precursors. (a) XRD pattern of Co(OH)_2 precursor. (b) High-resolution XPS spectra in Co 2p (b) and O 1s (c) region for Co(OH)_2 precursor. (d) TEM image of Co(OH)_2 precursor.

Figure R7 | Characterizations for CoP support. (a) XRD pattern of CoP support. (b) High-resolution XPS spectra in Co 2p (b) and P 2p (c) region for CoP support. (d) TEM image of CoP support.

To prepare CoP support, the as-prepared Co(OH)_2 nanosheets (100 mg) and $\text{NaH}_2\text{PO}_2 \cdot \text{H}_2\text{O}$ (2 g) were put at two separate positions in a quartz boat with NaH_2PO_2 at the upstream side of the furnace. Subsequently, the temperature of the tube furnace was raised to 300 °C with a ramping rate of 5 °C/min and maintained at 300 °C for 60 min, and then naturally cooled to room temperature under the protection of Ar gas with a flow rate of 100 mL/min. Powder XRD pattern of the product is shown in Figure R7a and consistent with that of CoP standard (JCPDS # 29-0497), suggesting the successful synthesis of CoP support. In the high-resolution XPS of the product (Figure R7b), the typical signals of Co and P species in CoP were identified, further confirming the formation of CoP support (*J. Am. Chem. Soc.* 2016, 138, 14686-14693). TEM image (Figure R7c) clearly demonstrates the morphology of nanosheet for the CoP support. Overall, the CoP nanosheets as support were successfully synthesized. **The updated results can be found in our revised manuscript (Page 7) and Supplementary Information (Methods section).**

Comment 5. XPS results of benchmark Pt and Ir catalysts also added into Fig. S14 and compared

Response: Thanks for suggestions. Herein, the high-resolution XPS spectra in Pt 4f and Ir 4f region for commercial Pt/C (20 wt.%) and Ir/C (wt.%) catalysts were added into the Figure S14 of the initial manuscript. The updated results were shown in Figure R8. Compared with Pt/CoP, the XPS peaks of Pt 4f_{7/2} and Pt 4f_{5/2} for Pt₂Ir₁/CoP shift to the low binding energy, while its XPS peaks of Co 2p_{3/2} and Co 2p_{2/1} shift to the high binding energy. In addition, the XPS peaks of Pt 4f_{7/2} (71.2 eV) and Pt 4f_{5/2} (74.4 eV) for Pt₂Ir₁/CoP is close to the characters of the Pt⁰ (Pt 4f_{7/2} = 71.0 eV and Pt 4f_{5/2} = 74.3 eV) in Pt/C benchmark, while its XPS peaks of Ir 4f_{7/2} (60.6 eV) and Ir 4f_{5/2} (63.8 eV) is close to the characters of the Ir⁰ (Ir 4f_{7/2} = 60.8 eV and Ir 4f_{5/2} = 64.0 eV) in Ir/C benchmark. Above facts thus supported the limited charge-transfer from Pt₂Ir₁ to CoP in Pt₂Ir₁/CoP. **The updated results can be found in the Supplementary Information (Figure S21).**

Figure R8 | (a) High-resolution XPS spectra in Pt 4f region for Pt/CoP, Pt₂Ir₁/CoP catalysts and Pt/C benchmark. (b) High-resolution XPS spectra in Ir 4f region for Pt₂Ir₁/CoP catalysts and Pt/C benchmark. (c) High-resolution XPS spectra in Co 2p region for Pt/CoP and Pt₂Ir₁/CoP catalysts.

Comment 6. Line 239, the meaning of T is missing. Please explain it.

Response: Thanks for pointing this out. In the initial submission, the Nyquist plots were simulated by a double-parallel equivalent circuit model. The first parallel components (*T* and *R*₁) reflect the charge-transfer kinetics, in which *T* is related to the double layer capacitance and *R*₁ represents catalytic charge-transfer resistance (*J. Electroanal. Chem.*, 1997, 424, 141-151; *Electroch. Acta*, 1997, 42, 323-330; *J. Power Sources*, 2006, 158, 464-476). **We have updated the relevant discussions in our revised manuscript (Page 14 Line 15-17).**

Comment 7. As shown in Line 149-150, the XRD patterns of loaded metal could not be obtained due to the small size and low loading. Hence the physical characterization of loaded metals is somewhat insufficient, it is suggested to do further

characterization (e.g., XANES, EXAFS, and etc.)

Response: Thanks for suggestions. Theoretically, it is very easy for Pt and Ir to form alloys because their fundamental properties (atomic radius, configuration of extra-nuclear electron, relative atomic mass, etc) are very similar (see the periodic table of chemical element for details). In addition, the synthetic strategy in our initial manuscript (chemical reduction of metal-salt precursors) was widely accepted for realizing the loading of Pt-based alloy nanoparticles (*Chem. Commun.*, 2010, 46, 8401-8403; *Chem. Mater.*, 2009, 21, 3649-3654; *J. Am. Chem. Soc.*, 2014, 136, 1280-1283; *Langmuir*, 2010, 26, 2339-2345). Thus, it was expected that the Pt₂Ir₁ in Pt₂Ir₁/CoP catalysts should be in the form of an alloy.

Figure R9 | CO stripping voltammetry of the Pt/CoP, Pt₂Ir₁/CoP and Ir/CoP catalysts in 0.5 M H₂SO₄ at scan rate of 20 mV/s. The CO was pre-adsorbed at 0.05 V vs. RHE for 15 min in each experiment. The CO stripping currents were normalized to each other so that all of the peaks have the same magnitude to better compare the CO stripping peak characteristics.

Indeed, the XANES and EXAFS are the powerful tools to distinguish the alloy or the single metal in our case. However, we are very sorry that the COVID-19 makes the synchrotron radiation experiments extremely difficult. Although trying to reserve the facility, we have to wait for half of year to use the XANES and EXAFS based on synchrotron radiation light source.

In this case, to experimentally support this recognition, we have used a sensitive and reliable electrochemical method, the stripping of adsorbed carbon monoxide (CO) measurements (Figure R9), in which the CO stripping peak characteristics were validated as an indicator of the composition of Pt-based metal (*Chem. Commun.*, 2014, 50, 11558-11561; *J. Mater. Chem. A*, 2016, 4, 15400-15410). In the case of the Pt/CoP and Ir/CoP, the CO stripping peaks are seen to be centered at 0.83 and 0.93 V vs. RHE respectively, corresponding well to the previous reports (*J. Mater. Chem. A*, 2016, 4, 15400-15410; *Langmuir*, 1997, 13, 6713-6721). The much more positive CO

stripping peak and much larger full width at half maximum (FWHM) for Ir/CoP over those for Pt/CoP could be explained as arising from the higher desorption activation energy of CO from Ir (22 kcal/mol) compared to that from Pt (13 kcal/mol) (*J. Phys. Chem.*, 1988, 92, 5213-5221). If no Pt-Ir alloy is formed in the Pt₂Ir₁/CoP catalysts, the CO stripping voltammograms should show two clear peaks, centered at the potentials seen for the Pt/CoP and Ir/CoP catalysts alone (*J. Mater. Chem. A*, 2016, 4, 15400-15410). Surely, the single CO stripping peak at 0.88 V vs RHE with moderate FWHM for the Pt₂Ir₁/CoP demonstrate the formation of alloyed Pt₂Ir₁ with regulated overall CO adsorption strength rather than the formation of only individual Pt and Ir (*J. Mater. Chem. A*, 2016, 4, 15400-1541).

Combining the TEM, EDX and XPS results (Figure 3c and S5) for the Pt₂Ir₁/CoP catalysts in our initial manuscript, **we herein believe that the loaded nanoparticles in Pt₂Ir₁/CoP catalysts should be the alloyed Pt₂Ir₁. We have updated the relevant discussions in our revised manuscript (Page 8).**

Reviewer 3

The study combines in a useful way the two important phenomena, of H-spillover and of the interface charge accumulation controlled by the work function difference. Spillover used to be hot topic in the context of hydrogen storage and the authors should cite an early (prior to both [19-20]) theoretical study explaining its thermodynamics, by A.Singh et al. ACS Nano, 3, 1657, 2009. // Here, it is nice to see the simple descriptor--work function--is used again for the HER activity, I say “again” because it seems essentially similar to LUS (lowest unoccupied state) introduced as descriptor for HER by Y. Liu et al. Nature Energy, 6, 17127, DOI: 10.1038/nenergy.2017.127, 2017. Some parallels/comparison should be offered here (the authors may also read D. Chirdon and Y. Wu, Nature Energy, 6, 17132, 2017).

I think the main problem of this paper is that they do not really show there is H spillover in their system. They seem to assume that this is the case, and then study how to optimize the component materials so that the H would not be trapped at the interface. There could be other reasons (without involving the H spillover) for the observed improved performance: for example, the nanoparticle itself may get better. In fact, as stated in page 13, “the changes in the ΔG_H on site 1’ and site 2’ range from -0.20 eV to -0.39 eV and from -0.06 eV to -0.15 eV...” this means that by alloying the particle itself gets better. There seems to be no need to involve the interface and further H spill over. The authors must try best to clarify strengthen this part, factual evidence of logical support for spillover action, in a good revision.

Response: We thank the reviewer for raising the useful comments on our manuscript. What we presented in this manuscript is *a fundamental understanding on hydrogen spillover phenomenon of metal-supported HER electrocatalysts and the importance of work function difference between metal and support ($\Delta\phi$) on the hydrogen spillover as well as HER activity for the hydrogen-spillover-based binary (HSBB) catalyst of Pt₂Ir₁/CoP*. The nature of the $\Delta\Phi$ and its contribution to HER performance are mainly embodied in two aspects. (a) *$\Delta\Phi$ affects the interfacial charge accumulation*. Replacing Pt by Pt₂Ir₁ alloy in Pt/CoP system will offset the intrinsic $\Delta\Phi$ between two components of the interface and thus restrain the interfacial charge flow, resulting in the reduced charge accumulation at the interface. This then enables the interfacial sites with the thermo-neutral hydrogen adsorption to become the

mediators for the energetically favorable interfacial hydrogen spillover. With these functions enabled, the energy barrier for interfacial hydrogen spillover will be significantly reduced and a hydrogen spillover channel of $\text{Pt}_2\text{Ir}_1 \rightarrow \text{interface} \rightarrow \text{CoP}$ is formed. **(b) $\Delta\Phi$ induces the surface charge relocation.** The small $\Delta\Phi$ of binary components in $\text{Pt}_2\text{Ir}_1/\text{CoP}$ results in the charge redistribution, eventually forming an electron enrichment region on Pt_2Ir_1 instead of the interface. This character endows the enhanced proton adsorption on the alloyed Pt_2Ir_1 sites, which is also beneficial for the interfacial hydrogen spillover. For these reasons, a high performance binary $\text{Pt}_2\text{Ir}_1/\text{CoP}$ HER electrocatalysts are realized with a close work function, which have displayed strong hydrogen adsorption on Pt_2Ir_1 , energetically favorable hydrogen spillover ($\text{PtIr} \rightarrow \text{interface} \rightarrow \text{CoP}$), and the efficient hydrogen desorption on CoP.

The reviewer mentioned the previous theoretically study on the hydrogen spillover phenomenon (*ACS Nano*, 2009, 3, 1657) of metal-support catalysts, which proposed that the metal should have a large enough hydrogen chemical potential relative to the support to enable the thermodynamically favorable hydrogen spillover. Inspired by this, it was realized that the $\text{Pt}_2\text{Ir}_1/\text{CoP}$ catalysts in our case should be essentially similar. These alloyed Pt_2Ir_1 sites with enriched electron density were endowed with enhanced proton adsorption. Hence, the hydrogen chemical potential on Pt_2Ir_1 should be significantly increased, thermodynamically facilitating the interfacial hydrogen spillover from Pt_2Ir_1 to CoP. This previous theoretically work provides a strong support on our study and helps us better explain the energetically favorable hydrogen spillover in $\text{Pt}_2\text{Ir}_1/\text{CoP}$ catalysts. **We have added the relevant discussions in Page 20-21 and cited this paper as Ref [73] in our revised manuscript.**

Most importantly, reviewer raised the concern on **the existence of hydrogen spillover phenomenon in our $\text{Pt}_2\text{Ir}_1/\text{CoP}$ catalysts.** Indeed, the catalytic system in our work is complicated, in which many parameters could affect the catalytic activity. In this revised manuscript, we would like to supply and reorganize more experimental results as well as presentation and to provide the solid experimental evidences and demonstrate the hydrogen spillover process in our study.

(1) *Explicitation of catalytic contributions: the catalytic performance of various control catalysts.*

Except for the contributions of hydrogen spillover, the HER activity improvements for the $\text{Pt}_2\text{Ir}_1/\text{CoP}$ catalysts may be also related to the contributions of

Pt₂Ir₁ or CoP itself. This view is widely accepted while revisiting the previous efforts on the metal-supported HER electrocatalysts. For instance, Baek *et.al* (*Adv. Mater.* 2018, 30, 1805606) reported the superior HER activity of catalysts of encapsulating Ir nanoparticles inside a cage-like organic network (Ir@CON), which took benefits from the intrinsically decent HER activity of Ir species and further optimization for its spatial structures by CON. Yang *et. al* (*Energy Environ. Sci.*,2020, 13, 3110) reported the catalysts of introducing single-atom Pt dopant into the Co₂P catalysts (Pt-Co₂P) significantly optimized the electronic structures and thereby HER process of Co₂P, affording the superior HER performance. Clearly, the catalysts mentioned above have either high metal loading (Ir@CON, 22 wt.%) or small metal size (Pt-Co₂P, atomic scale). These characters were beneficial to achieve abundant HER catalytic sites on metal as well as strong electronic metal-support interaction, which were the preconditions of achieving the profound contributions of metal or support itself for the overall HER activity. Considering the low total Pt₂Ir₁ loading (1.0 wt.%) and the large Pt₂Ir₁ size (~ 1.6 nm) in Pt₂Ir₁/CoP catalysts, it was expected that Pt₂Ir₁ or CoP itself in Pt₂Ir₁/CoP catalysts would not contribute profoundly to the overall HER activity. To experimentally examine the roles of Pt₂Ir₁ or CoP and figure out the nature behind the HER activity improvements, various control experiments have been designed and carried out.

(a) Initially, we have supplemented the control catalysts by loading Pt and Pt₂Ir₁ nanoparticles on reduced graphene oxide (Pt/rGO and Pt₂Ir₁/rGO) through the similar approach in the initial submission to examine the contributions of metals for such high catalytic performance of Pt₂Ir₁/CoP. It was found that the chemical and morphological characters of the loaded metals in Pt/rGO and Pt₂Ir₁/rGO, especially the loading (~ 1.0 wt.%) and size (~ 1.58 nm) were similar (Figure R1 and R2) to the catalysts of Pt/CoP and Pt₂Ir₁/CoP, therefore excluding their influences on the catalytic performance for various catalysts.

Figure R1 | Characterizations for the Pt/rGO catalysts. (a) High-resolution X-ray photoelectron spectra (XPS) spectrum in Pt 4f region for Pt/rGO catalysts. (b) Transmission electron microscopy (TEM) images for the Pt/rGO catalysts. (c) Dark-field scanning transmission electron microscope (STEM) images and (d) energy-dispersive X-ray (EDX) mapping for the Pt/rGO catalysts. (e) Inductively-coupled plasma mass spectrometry (ICP-MS) analysis for the Pt/rGO catalysts.

Figure R2 | Characterizations for the Pt₂Ir₁/rGO catalysts. (a) High-resolution XPS spectra in Pt 4f and Ir 4f regions. (b) TEM images. (c) STEM images, EDX mapping and line scan. (d) ICP-MS analysis.

Figure R3 | Catalytic contribution analysis. (a) HER performance comparison of Pt/CoP, Pt₂Ir₁/CoP, Pt/rGO and Pt₂Ir₁/rGO catalysts. (b) HER performance comparison of Pt/CoP and Pt₂Ir₁/CoP catalysts in the presence or absence of the thiocyanate ions (SCN⁻) probe.

In Figure R3a, the rGO was catalytically inert for HER, consistent with the previous reports (*J. Am. Chem. Soc.* 2011, 133, 7296–7299; *Adv. Funct. Mater.* 2016, 26, 6785–6796). Thus, the apparent HER activity of Pt/rGO and Pt₂Ir₁/rGO could effectively embody the catalytic contributions of the loaded metals themselves. Experimentally, the Pt₂Ir₁/rGO catalysts showed far higher overpotential of 193 mV to reach 20 mA/cm² (η_{20}) and much larger Tafel slope of 86.2 mV/dec, in comparison with those of Pt₂Ir₁/CoP (η_{20} = 7 mV and Tafel slope = 25.2 mV/dec) and even CoP (η_{20} = 156 mV and Tafel slope = 103 mV/dec), suggesting the non-dominating contributions of Pt₂Ir₁ in Pt₂Ir₁/CoP. In addition, the η_{20} and Tafel slope of Pt₂Ir₁/rGO were slightly less than that of Pt/rGO (η_{20} = 166 mV and Tafel slope = 86.2 mV/dec), **further excluding the case that Pt₂Ir₁ itself in Pt₂Ir₁/CoP catalysts enhanced and dominated the HER activity improvements in our catalytic system.** These results could also correspond to the DFT calculations in the initial manuscript (Figure 6a). As the reviewer noticed, compared with Pt/CoP model, the Pt₂Ir₁/CoP showed the changes in the ΔG_{H} on Pt₂Ir₁ itself (site 1': -0.20 eV → -0.39 eV; site 2': -0.06 eV → -0.15 eV), indicating the stronger hydrogen adsorption at the Pt₂Ir₁ sites. When considering the solo HER on Pt₂Ir₁ itself, the stronger hydrogen adsorption enables faster proton supply for the reaction, however, on the other side, this leads to weaker hydrogen desorption and slower release of active sites, still limiting the overall HER rate (*Science*, 2017, 355, eaad4998). From this view, the less HER activity of Pt₂Ir₁/rGO relative to Pt/rGO is predictable.

(b) To investigate the catalytic contributions of the CoP itself, we have incorporated thiocyanate ions (SCN⁻), which are known to block and deactivate the Pt

and Ir sites under acidic conditions (*Adv. Mater.* 2018, 30, 1805606; *Energy Environ. Sci.*, 2020, 13, 4921-4929). In Figure R3b, the CoP-like HER activity for SCN-Pt₂Ir₁/CoP ($\eta_{20} = 149$ mV and Tafel slope = 107.1 mV/dec) provided the convincing experimental evidences that the catalytic contribution of CoP itself was non-dominating. In addition, the HER activity of SCN-Pt₂Ir₁/CoP was very close to that of SCN-Pt/CoP ($\eta_{20} = 151$ mV and Tafel slope = 100.1 mV/dec), **excluding the dominated contribution of CoP itself in Pt₂Ir₁/CoP catalysts for the HER activity improvements.**

Above control experiments provided strong evidences to prove the significantly enhanced HER activity of Pt₂Ir₁/CoP catalysts was not resulted from Pt₂Ir₁ or CoP itself. **Naturally, the hydrogen spillover phenomenon should be the origin for such high HER activity of Pt₂Ir₁/CoP catalysts. For a better understanding, the updated discussions can be found in our revised manuscript (Page 11-14 and Page 19).**

(2). Evidences of hydrogen spillover phenomenon: the *operando* electrochemical impedance spectra (EIS) and cyclic voltammetry (CV) investigations

The previous efforts proposed that hydrogen spillover was highly relevant to the properties of supports in heterogeneous catalysis and the utilization of the *operando* spectroscopy (such as in-situ XAFS) on supports could confirm the hydrogen spillover phenomenon (*Nature*, 2017, 541, 68-71; *Nat. Nanotech.* 2020, 15, 848-853). Generally, supports in heterogeneous catalysts are usually unable to realize the dissociative adsorption of hydrogen to form hydrogen intermediates under the operations. Thus, monitoring the hydrogen intermediates on supports by *operando* spectroscopy can provide the convincing evidences on the occurrence of hydrogen spillover from metal to support. Unlike those cases, in our case, CoP could also adsorb hydrogen and thus bring the ambiguity whether the hydrogen intermediates on CoP originate from hydrogen adsorption on itself or hydrogen spillover. In addition, the complex electrolyte environment (containing H₂O, H⁺, H₃O⁺ and SO₄²⁻) in electrocatalytic HER goes against the utilization of common *operando* spectroscopy. Thus, the use of common *operando* spectroscopies is impossible for our study.

Fortunately, inspired by the previous efforts, it was expected that the hydrogen adsorption and desorption behavior on CoP support in Pt₂Ir₁/CoP catalysts should be

substantially different if hydrogen spillover indeed exists.

(a) Hydrogen adsorption behavior. The *operando* EIS investigations were carried out on bare CoP, Pt/CoP, Pt₂Ir₁/CoP, Pt/rGO, Pt₂Ir₁/rGO, SCN-Pt/CoP and SCN-Pt₂Ir₁/CoP catalysts at different overpotentials (*Energy Environ. Sci.*, 2019, 12, 2298-2304). The recorded Nyquist plots were simulated by a double-parallel equivalent circuit model (Figure R4 and Table R3). Following a previous recognition (*Energy Environ. Sci.*, 2019, 12, 2298-2304), the second parallel components (C_ϕ and R_2) reflect the hydrogen adsorption behavior on catalyst surface, where C_ϕ and R_2 represent the hydrogen adsorption pseudo-capacitance and resistance, respectively.

Figure R4 | EIS analysis. Nyquist plots for (a) bare CoP, (b) Pt/CoP, (c) Pt₂Ir₁/CoP, (d) Pt/rGO, (e) Pt₂Ir₁/rGO, (f) SCN-Pt/CoP and (g) SCN-Pt₂Ir₁/CoP catalysts in 0.5 M H₂SO₄ at various HER overpotentials. Zoom-in parts were correspondingly presented as inset. The scattered symbols represent the experimental results, and the solid lines are simulation fitted results. The inset also shows the equivalent circuit for the simulation. The fitted parameters are summarized in Table R1.

Table R1 | The fitted parameters of the EIS data of bare CoP, Pt/CoP, Pt₂Ir₁/CoP, Pt/rGO, Pt₂Ir₁/rGO, SCN-Pt/CoP and SCN-Pt₂Ir₁/CoP for HER.

Catalysts	η [mV]	R_s [Ω]	T [F s ⁻¹]	R_1 [Ω]	n_1	R_2 [Ω]	C_ϕ [F]
CoP	0	3.58	0.0041	21.1	0.86	9120	0.0018
	-10	3.68	0.0039	21.0	0.90	7762	0.0024
	-20	3.63	0.0043	20.9	0.87	6310	0.0026
	-30	3.61	0.0042	20.8	0.83	5370	0.0032
	-40	3.64	0.0045	20.4	0.81	4266	0.004
	-50	3.59	0.0044	19.8	0.90	3162	0.007
	-60	3.65	0.0042	17.7	0.91	2512	0.010
	-70	3.61	0.0044	16.7	0.85	2138	0.0132
	-80	3.66	0.0042	15.8	0.86	1549	0.0145
	-90	3.57	0.0041	15.0	0.88	1191	0.0172
	-100	3.61	0.0045	14.6	0.83	879	0.0203
	-110	3.63	0.0046	13.9	0.89	616	0.0224
	-120	3.60	0.0038	12.8	0.92	340	0.024
	-130	3.54	0.0045	12.0	0.91	160	0.026
-140	3.61	0.0041	11.4	0.81	–	–	
Pt/CoP	0	3.58	0.0058	24.1	0.82	8912	0.0015
	-10	3.62	0.0062	24.0	0.91	6310	0.0025

	-20	3.57	0.0061	23.7	0.87	4786	0.004
	-30	3.56	0.0058	24.0	0.86	3715	0.0063
	-40	3.55	0.006	23.1	0.9	3020	0.0093
	-50	3.51	0.0054	22.8	0.91	2042	0.013
	-60	3.56	0.0059	21.0	0.83	1514	0.016
	-70	3.50	0.0064	20.7	0.88	1000	0.017
	-80	3.51	0.0061	20.4	0.87	735	0.018
	-90	3.49	0.006	19.8	0.93	588	0.019
	-100	3.63	0.0064	19.2	0.91	410	0.02
	-110	3.62	0.0058	18.8	0.90	299	0.021
	-120	3.55	0.0051	18.3	0.87	201	0.022
	-130	3.59	0.0065	18.0	0.86	109	0.0222
	-140	3.60	0.0062	17.3	0.82	–	–
Pt/C	0	3.42	0.0025	35.2	0.90	5623	0.0001
	-10	3.51	0.0031	34.7	0.81	3630	0.0002
	-20	3.50	0.0027	34.3	0.83	2570	0.0007
	-30	3.55	0.0039	34.2	0.83	2089	0.0016
	-40	3.53	0.0033	33.8	0.85	1380	0.0027
	-50	3.51	0.0031	33.3	0.82	1023	0.0037
	-60	3.49	0.0028	32.9	0.85	676	0.0047
	-70	3.46	0.0037	32.1	0.92	389	0.0055
	-80	3.51	0.0036	31.6	0.91	239	0.0061
	-90	3.42	0.0035	31.0	0.90	125	0.0065
	-100	3.44	0.0038	30.4	0.92	–	–
SCN-Pt/CoP	0	3.61	0.0055	25.1	0.91	8912	0.0018
	-10	3.58	0.0051	24.6	0.85	7244	0.0018
	-20	3.62	0.0057	24.2	0.86	5495	0.0020
	-30	3.60	0.0049	24.1	0.86	4466	0.0028
	-40	3.65	0.0053	23.7	0.88	3467	0.0033
	-50	3.62	0.0051	23.2	0.87	2570	0.0054
	-60	3.58	0.0058	22.8	0.81	1995	0.0097
	-70	3.54	0.0057	22.2	0.89	1479	0.014
	-80	3.55	0.0056	21.7	0.88	1149	0.015
	-90	3.66	0.0054	21.2	0.92	891	0.018
	-100	3.67	0.0049	20.5	0.81	660	0.021
-110	3.60	0.0052	20.0	0.80	457	0.023	
-120	3.59	0.0053	19.6	0.83	281	0.025	
-130	3.61	0.0058	19.2	0.86	141	0.026	
-140	3.67	0.0056	18.7	0.80	–	–	
Pt₂Ir₁/CoP	0	3.52	0.0060	24.9	0.90	4365.2	0.007
	-10	3.51	0.0054	24.2	0.92	2512	0.019
	-20	3.55	0.0055	23.6	0.81	1621.8	0.051
	-30	3.56	0.0056	23.5	0.80	891.3	0.066
	-40	3.61	0.0057	23.0	0.85	512.9	0.081
	-50	3.62	0.0059	22.6	0.82	263.2	0.088
	-60	3.49	0.0062	22.0	0.81	77.6	0.096
-70	3.50	0.0061	21.4	0.83	–	–	
Pt₂Ir₁/C	0	3.51	0.0035	36.4	0.91	4677	0.0001
	-10	3.53	0.0030	35.9	0.82	3020	0.0007
	-20	3.54	0.0037	35.5	0.82	2188	0.0024
	-30	3.45	0.0029	35.4	0.82	1259	0.0037
	-40	3.46	0.0023	35.0	0.86	871	0.0048
	-50	3.55	0.0035	34.5	0.83	646	0.0058
	-60	3.57	0.0038	34.1	0.83	389	0.0065
	-70	3.53	0.0027	33.9	0.90	246	0.0069
	-80	3.50	0.0034	32.8	0.93	100	0.0072
-90	3.47	0.0036	32.2	0.91	–	–	

	0	3.42	0.0045	26.3	0.81	9549	0.0010
	-10	3.51	0.0041	25.8	0.95	8511	0.0012
	-20	3.52	0.0047	25.5	0.82	6918	0.0017
	-30	3.50	0.0039	25.3	0.81	5888	0.0020
	-40	3.55	0.0043	24.9	0.80	4786	0.0038
	-50	3.52	0.0055	24.4	0.82	3631	0.0064
	-60	3.49	0.0049	23.9	0.83	2951	0.011
SCN-Pt₂Ir₁/CoP	-70	3.44	0.0047	23.4	0.86	2290	0.013
	-80	3.45	0.0046	22.9	0.82	1659	0.016
	-90	3.56	0.0044	22.4	0.82	1258	0.019
	-100	3.57	0.0047	21.7	0.91	851	0.021
	-110	3.50	0.0051	21.2	0.90	616	0.024
	-120	3.49	0.0050	20.8	0.93	338	0.025
	-130	3.51	0.0052	20.2	0.85	169	0.026
	-140	3.57	0.0051	19.6	0.90	–	–

As shown in Figure R5, the integration of C_ϕ vs. η profiles gives the hydrogen adsorption charge (Q_H) on catalyst surfaces during HER. The equally small Q_H values of Pt/rGO ($Q_H[\text{Pt}/\text{rGO}] = 285 \mu\text{C}$) and Pt₂Ir₁/rGO ($Q_H[\text{Pt}_2\text{Ir}_1/\text{rGO}] = 340 \mu\text{C}$) while similar Q_H values of bare CoP ($Q_H[\text{CoP}] = 1547 \mu\text{C}$), SCN-Pt/CoP ($Q_H[\text{SCN-Pt}/\text{CoP}] = 1549 \mu\text{C}$) and SCN-Pt₂Ir₁/CoP ($Q_H[\text{SCN-Pt}_2\text{Ir}_1/\text{CoP}] = 1576 \mu\text{C}$) further supported that the enhancement of Pt₂Ir₁ or CoP itself in Pt₂Ir₁/CoP catalysts was too slight to dominate its hydrogen adsorption behavior and HER activity improvements. Naturally, the almost non-increased amount of adsorbed hydrogen on CoP in Pt/CoP ($Q_H[\text{Pt}/\text{CoP}] - Q_H[\text{Pt}/\text{rGO}] = 1515 \mu\text{C}$) compared with that on bare CoP ($Q_H[\text{CoP}] = 1547 \mu\text{C}$) rationalized the limited hydrogen spillover from Pt to CoP. Comparatively, the exponentially increased amount of adsorbed hydrogen on CoP in Pt₂Ir₁/CoP ($Q_H[\text{Pt}_2\text{Ir}_1/\text{CoP}] - Q_H[\text{Pt}_2\text{Ir}_1/\text{rGO}] = 3225 \mu\text{C}$) compared to that in Pt/CoP ($Q_H[\text{Pt}/\text{CoP}] - Q_H[\text{Pt}/\text{rGO}] = 1515 \mu\text{C}$) strongly suggested **the existence of hydrogen spillover from Pt₂Ir₁ to CoP as a new effortless acquisition path for the hydrogen adsorption of CoP.**

Figure R5 | Plots of C_ϕ vs. η for bare CoP, Pt/CoP, Pt₂Ir₁/CoP, Pt/rGO, Pt₂Ir₁/rGO, SCN-Pt/CoP and Pt₂Ir₁/CoP during HER in Ar-saturated 0.5 M H₂SO₄.

Figure R6 | EIS-derived Tafel plots for bare CoP, Pt/CoP and Pt₂Ir₁/CoP catalysts obtained from the hydrogen adsorption resistance R_2 . The solid lines represent linear fitting.

In this way, the corresponding hydrogen adsorption kinetics should also change. Considering the potential-dependent R_2 for all catalysts, it is rational to quantify their hydrogen adsorption kinetics *via* plotting $\log R_2$ vs. overpotential and calculating the EIS-derived Tafel slopes by virtue of the Ohm's law (*J. Power Sources* 2006, 158, 464-476). As shown in Figure R6, the similar EIS-derived Tafel slope of Pt/CoP compared with that of bare CoP suggests its unaltered hydrogen adsorption kinetics. Hence, Pt/CoP showed the individual hydrogen adsorption on respective Pt and CoP, supporting the limited hydrogen spillover from Pt to CoP due to the sluggish spillover kinetics. Comparatively, the significantly declined EIS-derived Tafel slope for Pt₂Ir₁/CoP indicates an accelerated hydrogen adsorption kinetics. **Such phenomenon revealed that the intrinsically insufficient hydrogen adsorption on CoP was replaced by a successful hydrogen spillover from Pt₂Ir₁ to CoP as a new faster pathway for hydrogen adsorption owing to the profoundly enhanced spillover kinetics (better than the kinetics of solo hydrogen adsorption on CoP).**

(b) Hydrogen desorption behavior. To figure out the hydrogen desorption behavior, the *operando* CV investigations were performed on bare CoP, Pt/CoP, Pt₂Ir₁/CoP, Pt/rGO, Pt₂Ir₁/rGO, SCN-Pt/CoP and SCN-Pt₂Ir₁/CoP catalysts and monitored their hydrogen desorption peak during CV scanning in the double layer region (*Angew. Chem. Int. Edit.* 2019, 58, 16038-16042; *J. Am. Chem. Soc.* 2009, 131, 14756-14760; *J. Mater. Chem. A* 2014, 2, 3954-3960; *Int. J. Hydrogen Energy* 2018, 43, 1251-1260). As shown in Figure R7, the CV curves shows that the intensities of the hydrogen desorption peaks for Pt/rGO and Pt₂Ir₁/rGO are equally weak. Similar characteristics were also found on the hydrogen desorption peaks of bare CoP,

SCN-Pt/CoP and SCN-Pt₂Ir₁/CoP. Above facts further supported that the enhancement of Pt₂Ir₁ or CoP itself in Pt₂Ir₁/CoP catalysts was too slight to dominate its hydrogen desorption behavior and thereby HER activity improvements. Naturally, the similar hydrogen desorption peak of Pt/CoP compared with that of bare CoP suggests its almost non-increased amount of desorbed hydrogen, corresponding to the limited hydrogen spillover from Pt to CoP and thus the lack of abundant spillovered hydrogen for desorption. **In contrast, the significantly stronger hydrogen desorption peak of Pt₂Ir₁/CoP compared to that of Pt/CoP strongly indicated the existence of hydrogen spillover from Pt₂Ir₁ to CoP and thus the excess spillovered hydrogen on CoP for efficient desorption.**

Figure R7 | CV curves of bare CoP, Pt/CoP, Pt₂Ir₁/CoP, Pt/rGO, Pt₂Ir₁/rGO, SCN-Pt/CoP and SCN-Pt₂Ir₁/CoP catalysts in Ar-saturated 0.5 M H₂SO₄ with a scan rate of 50 mV/s.

Above facts could also be supported by analyzing the corresponding hydrogen desorption kinetics. The CV curves of bare CoP, Pt/CoP and Pt₂Ir₁/CoP catalysts show the hydrogen desorption peak shift depending on the scan rate because the current in the catalysts takes more time to respond to the applied potential (*J. Phys. Chem. C* 2011, 115, 11880-11886). Thus, it is rational to quantify their hydrogen desorption kinetics *via* plotting hydrogen desorption peak position *vs.* scan rate and comparing the fitted slopes (Figure R8a-8c). As shown in Figure R8d, the similar fitted slope of Pt/CoP compared with that of bare CoP suggests its unaltered hydrogen desorption kinetics. Comparatively, the significantly reduced slope for Pt₂Ir₁/CoP suggests its drastically accelerated hydrogen desorption kinetics. It was reported that the hydrogen desorption kinetics for metal-support electrocatalysts could be effectively accelerated by hydrogen spillover effect (*Angew. Chem. Int. Edit.* 2019, 58, 16038-16042).

Therefore, the unaltered kinetics of the hydrogen desorption for Pt/CoP should be related to the limited hydrogen spillover from Pt to CoP, while the faster kinetics of the hydrogen desorption for Pt₂Ir₁/CoP should be originated from the efficient hydrogen spillover from Pt₂Ir₁ to CoP.

Figure R8 | Capacitance vs. voltage profiles obtained from cyclic voltammograms of (a) bare CoP, (b) Pt/CoP and (c) Pt₂Ir₁/CoP catalysts with the scan rate from 50 to 850 mV/s in Ar-saturated 0.5 M H₂SO₄. (d) Plots of hydrogen desorption peak position vs. scan rate for bare CoP, Pt/CoP and Pt₂Ir₁/CoP catalysts. The solid lines represent linear fitting.

(c) **Consistence of theoretically calculated and experimentally measured Tafel slopes.** Referring to the previous reports (*Nat. Commun.* 2016, 7, 12272; *Adv. Funct. Mater.* 2017, 27, 1700359), the HER pathway of Pt₂Ir₁/CoP was described by the following equations:

The reaction velocity of hydrogen evolution could be written as $r = k_3\theta_{\text{CoP}-\text{H}^*}C_{\text{H}^+}$, where r is the reaction rate; k is the rate constant; θ is the hydrogen coverage of on active sites; and C_{H^+} is the concentration of hydrogen ion.

In the steady state,

$$\frac{d\theta_{CoP-H^*}}{dt} = k_2\theta_{Pt_2Ir_1-H^*}(1-\theta_{CoP-H^*}) - k_{-2}\theta_{CoP-H^*}(1-\theta_{Pt_2Ir_1-H^*}) - k_3\theta_{CoP-H^*}C_{H^+}$$

$$\frac{d\theta_{Pt_2Ir_1-H^*}}{dt} = k_1(1-\theta_{Pt_2Ir_1-H^*})C_{H^+} - k_{-1}\theta_{Pt_2Ir_1-H^*} - k_{-2}\theta_{Pt_2Ir_1-H^*}(1-\theta_{CoP-H^*}) + k_2\theta_{CoP-H^*}(1-\theta_{Pt_2Ir_1-H^*})$$

At the low overpotential,

$$\theta_{CoP-H^*} \approx \frac{k_2\theta_{Pt_2Ir_1-H^*}}{k_2\theta_{Pt_2Ir_1-H^*} + k_{-2} - k_{-2}\theta_{Pt_2Ir_1-H^*} + k_3C_{H^+}} \approx \frac{k_2}{k_{-2}}\theta_{Pt_2Ir_1-H^*}e^{-\frac{F\Delta\phi}{RT}}$$

$$\theta_{Pt_2Ir_1-H^*} \approx \frac{k_1C_{H^+} + k_{-2}\theta_{CoP-H^*}}{k_1C_{H^+} + k_{-1} + k_2 + k_{-2}\theta_{CoP-H^*} - k_2\theta_{CoP-H^*}} \approx \frac{k_1}{k_{-1}}C_{H^+}e^{-\frac{F\Delta\phi}{RT}}$$

Thus,

$$r \approx k_3\theta_{CoP-H^*}C_{H^+} = \frac{k_1k_2k_3}{k_{-1}k_{-2}}C_{H^+}^2e^{-\frac{(2+\alpha)F\Delta\phi}{RT}} \quad (4)$$

And,

$$-j = Fr = \frac{k_1k_2k_3}{k_{-1}k_{-2}}FC_{H^+}^2e^{-\frac{(2+\alpha)F\Delta\phi}{RT}}$$

$$\lg(-j) = Constant + 2\lg C_{H^+} - \frac{(2+\alpha)F}{2.303RT}\Delta\phi$$

Therefore, the calculated Tafel slope for Pt₂Ir₁/CoP catalysts is: $\frac{(2+\alpha)F}{2.303RT} = 0.023$

V/dec (assuming $\alpha = 0.5$, F is the Faraday constant, R the Rydberg gas constant and T the absolute temperature). The coincidence between this theoretical value and experimental observation (Figure R3a, 25.2 mV/dec) for Pt₂Ir₁/CoP clearly confirmed this hydrogen-spillover-based HER pathway.

(d) pH- and temperature-dependent HER performance. Such HER pathway could be further verified by investigating the pH-dependent relation of HER (Figure R9a and 9b). In this way, the reaction order of Pt₂Ir₁/CoP was experimentally determined to be 1.98, which was also in accord with the theoretical value of 2 (Equation 4). Moreover, the temperature-dependent relation of HER for Pt₂Ir₁/CoP catalysts was also investigated (Figure R9c and R9d). A linear relationship between $\log j_0$ with $1/T$ is exhibited in a semi-logarithmic plot and then electrochemical activation energies could be calculated according to the Arrhenius equation ($\log j_0 = \log(FK_c) - \Delta G_0/2.303RT$, where R is the gas constant, ΔG_0 is the apparent activation energy, F is the Faraday constant). The calculated value of ΔG_0 for Pt₂Ir₁/CoP is 20.9

kJ/mol. Such low activation energy is beneficial to HER process. All these characters of $\text{Pt}_2\text{Ir}_1/\text{CoP}$ corresponded well to those of the previously reported hydrogen-spillover-based HER electrocatalysts in acid media (*Nat. Commun.* 2016, 7, 12272; *Adv. Funct. Mater.* 2017, 27, 1700359) and strongly supported its successful hydrogen spillover process and profound contributions to the HER activity.

Figure R9 | (a) Tafel curves of $\text{Pt}_2\text{Ir}_1/\text{CoP}$ catalysts in Ar-saturated H_2SO_4 with pH ranging from 0 to 0.68. (b) plots of $\log j$ at -0.05 V (vs. RHE) vs. pH for $\text{Pt}_2\text{Ir}_1/\text{CoP}$ catalysts. (c) Tafel curves of $\text{Pt}_2\text{Ir}_1/\text{CoP}$ catalysts in Ar-saturated 0.5 M H_2SO_4 at different temperatures ranging from 298 to 338 K. (d) Typical Arrhenius plots for $\text{Pt}_2\text{Ir}_1/\text{CoP}$ catalysts. The solid lines represent linear fitting.

Depending on the above experimental results, a clear logical clue came into being. *Such high HER performance of $\text{Pt}_2\text{Ir}_1/\text{CoP}$ catalysts should be derived from the hydrogen spillover phenomenon rather than Pt_2Ir_1 or CoP itself. Naturally, the unique observations on hydrogen adsorption/desorption behavior and apparent reaction kinetic performance for $\text{Pt}_2\text{Ir}_1/\text{CoP}$ really show there is hydrogen spillover in this catalytic system.* Hence, *it can be convincing that the $\Delta\phi$ like a descriptor shows the predictive power for the hydrogen spillover phenomenon and thus the HER activity of the PtM/CoP hydrogen spillover based electrocatalysts.* For a better understanding, we have updated the relative discussion in our revised

manuscript (Page 14-18).

Similar to the $\Delta\phi$, the work function related descriptor for the HER activity of the transition-metal dichalcogenides (MX_2) has been reported by Y. Liu *et al.* (*Nat. Energy* 2017, 2, 17127). The researchers identify that the energy at or near the lowest unoccupied states (ϵ_{LUS}) for MX_2 is the key determinant of hydrogen adsorption strength on their basal plane as the hydrogen adsorption on MX_2 leaves the profile of the electronic density of states largely intact, with complete charge transfer from the adsorbate to the catalyst. Thus, the ϵ_{LUS} can be regarded as the electronic descriptor to reflect the hydrogen adsorption and desorption rate on MX_2 themselves and thus their HER activity. Essentially, the ϵ_{LUS} is similar to the classic volcano theory in which hydrogen adsorption free energy (ΔG_{H}) on catalyst surfaces is used as an indicator of the catalytic efficiency (*Science*, 2017, 355, eaad4998). In this theory, if $\Delta G_{\text{H}} < 0$, the hydrogen adsorption is favorable but desorption is unfavorable. If $\Delta G_{\text{H}} > 0$, the hydrogen desorption is favorable but adsorption is unfavorable. Neither too strong nor too weak adsorption of active hydrogen species ($\Delta G_{\text{H}} \approx 0$) is recognized as the criterion for efficient HER electrocatalysts, since this state can evenly facilitate hydrogen adsorption and desorption. Analogously, by optimizing the ϵ_{LUS} , it is also expected to screen for the MX_2 with balanced hydrogen adsorption and desorption.

In contrast, the hydrogen adsorption and desorption in the HSBB catalysts of $\text{Pt}_2\text{Ir}_1/\text{CoP}$ were both energetically favorable as the deliberate design of utilizing metal with $\Delta G_{\text{H}} < 0$ for hydrogen adsorption and support with $\Delta G_{\text{H}} > 0$ for hydrogen desorption. Thus, the HER process was mainly determined by the interfacial hydrogen spillover. The more efficient hydrogen spillover was enabled, the better HER activity was achieved. Herein, the $\Delta\Phi$ is the key determinant for interfacial hydrogen spillover barrier as the $\Delta\Phi$ -determined interfacial charge accumulation/relocation and thus can reflect the HER activity of the HSBB catalysts. By optimizing the $\Delta\Phi$, it is expected to screen for the HSBB catalysts with efficient hydrogen spillover. ***Although the ϵ_{LUS} and $\Delta\Phi$ both delivered strong dependence on the HER activity, there is still the big differences in their targeted catalysts, motivation and mechanism (Table R2). We have added the relevant discussions in Page 6 and cited these works as Ref [31] and [32] in our revised manuscript.***

Table R2 | Comparison of ϵ_{LUS} and $\Delta\Phi$.

Descriptor	Targeted Catalyst	Mechanism	Motivation
ϵ_{LUS}	MX_2 catalysts	Hydrogen adsorption on MX_2 leaves the profile of the electronic density of states largely intact, with complete charge transfer from the adsorbate to the catalyst. Thus, the ϵ_{LUS} is the key determinant of hydrogen adsorption strength on MX_2 and used to reflect the hydrogen adsorption and desorption rate on MX_2 themselves and their HER activity.	Balanced hydrogen adsorption and desorption at the same sites on MX_2 themselves
$\Delta\Phi$	HSBB catalysts	Large difference in the Fermi energies of metal and support drives the interfacial charge flow until the system reaches an equilibrium, followed by the Schottky barrier formation and charge accumulation at the interface, thus strongly tapping proton at interface. In this case, the interfacial hydrogen spillover has to overcome a large energy barrier, leading to the unsatisfactory HER activity. Thus, the $\Delta\Phi$ is the key determinant of hydrogen spillover barrier and used to reflect the overall HER activity.	1. Efficient hydrogen spillover from metal to support2. Hydrogen adsorption and desorption at the different sites

REVIEWER COMMENTS

Reviewer #1 (Remarks to the Author):

The authors tried to design some experiments and make some discussion. But I do not think the updated data can provide solid evidences to support their conclusion on hydrogen spillover. The authors demonstrated that the larger hydrogen adsorption charge (QH) and the smaller EIS-derived Tafel slope of Pt₂Ir/CoP can account for superior HER activity. Meanwhile, they also tracked the peak position of hydrogen desorption. Although such change of hydrogen adsorption/desorption can account for the activity changes, it can arise from various reasons and can't be used to support hydrogen spillover theory. Besides, the authors synthesized rGO supported Pt and Pt₂Ir samples to confirm that the activity enhancement of Pt₂Ir/CoP was not derived from Ir or its alloying effect. The Ir/CoP should be also prepared for comparison.

Reviewer #3 (Remarks to the Author):

The revision is OK but the Response is so excessively long (43 pages, not to forget also 39 pages of the paper) that it reminds the "filibuster method" in some parliamentary proceedings. Anyways, the authors tried hard to address reviewers' concerns. I still do not see clear convincing evidence of the spillover process but I can accept it here as an overall plausible scenario of the process, OK. I do see and sympathize with the stronger reservations of the Reviewer 1. Yet from my perspective the paper overall is thorough and its publication can be useful, even though leaving some moot points debatable in the future. I can endorse its acceptance to Nat. Comm. at this point.

Author's Response to Reviewers

Reviewer 1

The authors tried to design some experiments and make some discussion. But I do not think the updated data can provide solid evidences to support their conclusion on hydrogen spillover. The authors demonstrated that the larger hydrogen adsorption charge (Q_H) and the smaller EIS-derived Tafel slope of Pt₂Ir/CoP can account for superior HER activity. Meanwhile, they also tracked the peak position of hydrogen desorption. Although such change of hydrogen adsorption/desorption can account for the activity changes, it can arise from various reasons and can't be used to support hydrogen spillover theory. Besides, the authors synthesized rGO supported Pt and Pt₂Ir samples to confirm that the activity enhancement of Pt₂Ir/CoP was not derived from Ir or its alloying effect. The Ir/CoP should be also prepared for comparison.

Response: We thank the reviewer for raising their concerns on our *operando* electrochemical impedance spectra (EIS) and cyclic voltammetry (CV) in confirming the presence of hydrogen spillover. We would like to address his/her comments.

Generally, hydrogen spillover phenomenon in heterogeneous catalysis takes place on the metal-supported heterogeneous catalysts. The supports are usually unable to realize the dissociative adsorption of hydrogen to form active hydrogen intermediates under operations. Thus, monitoring the spillovered hydrogen intermediates on supports by *operando* techniques (e.g. EXAFS, XANES, FT-IR and H₂-TPR etc) can experimentally provide solid evidence to confirm the presence of hydrogen spillover, especially for the thermal catalytic reactions (*Nature* 2017, 541, 68-71; *Nat. Nanotech.* 2020, 15, 848-853; *Chem. Rev.*, 2012, 112, 2714-2738).

Unlike those cases, in electrocatalytic hydrogen evolution reaction (HER), hydrogen could also be adsorbed on support (like CoP in our case) and thus bring the ambiguity whether the hydrogen intermediates on support originate from the hydrogen adsorption on itself or hydrogen spillover. In addition, the electrolyte environment (such as H₂O, H₃O⁺, SO₄²⁻, H⁺, OH⁻ and K⁺) during HER goes against the monitoring of the spillovered hydrogen on support. Faced with these common issues, the use of the state-of-the-art *operando* techniques (e.g. XAS, FTIR) to demonstrate the electrocatalytic hydrogen spillover faces great challenges. Thus, it is still lack of solid experimental evidences on the presence of hydrogen spillover in electrocatalytic HER. Generally, theoretical simulation and indirect evidences are

adopted as the state-of-the-art approach to understand the hydrogen spillover in electrocatalytic HER, such as the calculations of the corresponding thermodynamic and kinetic parameters (e.g. activation energy, energy difference, Tafel slope and reaction order) or investigations on the hydrogen spillover induced phase change of support (*Energy & Environmental Science*, 2019, 12, 2298-2304; *Nat. Commun.*, 2016, 7, 1-7; *Angew. Chem. Int. Ed.*, 2020, 59, 20423-20427; *Nano Energy*, 2020, 71, 104653).

To minimize the impact of the above common issues, what we enabled in this work is a strategy of utilizing *operando* electrochemical investigations on the hydrogen adsorption/desorption behavior on support, which is equivalent to the concept of monitoring the spillovered hydrogen on support in heterogeneous catalysis and experimentally support the existence of hydrogen spillover in our case.

As also stated by the Reviewer #3, **this strategy is an opportunity to provide solid experimental evidences on electrocatalytic hydrogen spillover phenomenon but leaving debate.** To be specific, such electrochemical investigations were performed in a standard three-electrode system, containing centimeter-sized reference, counter and working electrodes. The nano-sized catalysts were loaded on the working electrode. This situation means that such electrochemical investigations actually collect the overall informations about all sites on the catalyst surface. In other word, the behavior at the local sites of the catalysts, such as the hydrogen spillover at the interface in our case, is directly indistinguishable. Thus, utilizing electrochemical EIS and CV in our work are actually for detecting the results of hydrogen spillover, that is, the significantly facilitated hydrogen adsorption/desorption behavior on catalyst surface. It is undeniable that there are other possibilities, such as the contributions of Ir, PtIr alloying effects or the enhanced CoP as proposed by the Reviewer #1, that lead to the above results. Faced with these limitations, we further compared the proposed Pt₂Ir₁/CoP with various control catalysts of Pt/rGO, Pt₂Ir₁/rGO, Pt/CoP, Ir/CoP (See Figure R1-R4 and Table R1 for details), SCN-Pt/CoP and SCN-Pt₂Ir₁/CoP to exclude the above possibilities as much as possible. Then, the comparison of theoretically calculated and experimentally measured kinetic parameters along with DFT calculations further support the existence of hydrogen spillover. In a summary, we have performed various experiments and calculations to support the occurrence of hydrogen spillover:

a). Comparison of various control catalysts with Pt₂Ir₁/CoP (Figure S12, S19

and Figure 6)

- b). Hydrogen adsorption behavior analysis (Figure 6a and 6b);*
- c). Hydrogen desorption behavior analysis (Figure 6c and 6d);*
- d). Comparison of theoretically calculated and experimentally measured Tafel slopes (Figure 4b and Equation S5);*
- e). pH- and temperature-dependent HER performance (Figure 6e and 6f);*
- f). Interfacial charge dilution and hydrogen spillover channel formation revealed by DFT calculations (Figure 7).*

Although these evidences are indirect, all above results demonstrate the hydrogen spillover phenomenon as the most likely reason for the significantly improved HER performance herein. We have done what we can do to demonstrate the electrocatalytic hydrogen spillover phenomenon in our case, which is indeed a significant advance compared with other studies on the HER electrocatalysts based on hydrogen spillover. Also, we provide a critical descriptor and a deep understanding on the design of highly performed HER electrocatalysts through hydrogen spillover.

We do agree with the importance of the direct evidences on hydrogen spillover proposed by the Reviewer #1. To obtain this goal, it could completely solve the above common issues on utilizing the state-of-the-art *operando* techniques for tracking the spillovered hydrogen. Unfortunately, with the current methodologies and technologies, this is almost impossible due to the inevitability of the existence of the adsorbed hydrogen on support as well as the H₂O in electrolyte. To address the Reviewer #1's concerns, the following aspects might be considered in the future:

a) Developing the transient imaging technology: If existing the efficient hydrogen spillover, large amounts of spillovered hydrogen may accumulate on support to form nanobubbles (*Proc. Natl. Acad. Sci.*, 2018, 115, 5878-5883). Thus, developing the transient imaging technology (like super-resolution fluorescence microscopy with suitable fluorescence dye molecules) to label the hydrogen nanobubbles and observe their nucleation, growth and migration will provide the solid evidences on hydrogen spillover. However, it requires a very high spatial resolution for this technique to identify the metals and supports.

b) Developing the nano-sized three-electrode electrochemical system: The limitations of our *operando* electrochemical investigations are derived from the much larger size of the three-electrode device compared to that of the catalysts. Developing the nano-sized three-electrode device might provide the possibility to investigate the

local electrochemical response on catalyst surface. In this way, a transient electrochemical response at the interface will be detected during hydrogen spillover.

c) Developing the special techniques to avoid the interference of solvents and electrolytes: The common issues of utilizing the state-of-the-art *operando* spectroscopy (EXFAS and FT-IR) to trace the spillovered hydrogen in current HER electrocatalysts are the interference of adsorbed hydrogen of support and various ion/molecule in ambience (H_2O , H_3O^+ , SO_4^{2-} , H^+ , OH^- and K^+) when using the common catalytic system (H_2SO_4 or KOH aqueous electrolyte). Thus, developing the special system may solve the above common issues without influencing the HER process of the catalysts, which lays the foundation for re-enabling the *operando* spectroscopy to trace the spillover hydrogen.

We will work in this regard and further promote the development of the novel HER electrocatalysts based on hydrogen spillover.

The Reviewer #1 also mentioned that the Ir/CoP catalysts should be also prepared for comparison. We thank the reviewer for this constructive suggestion. Herein, we have supplemented the control catalysts by loading Ir nanoparticles on CoP (Ir/CoP) through the similar approach to examine the contributions of Ir for such high catalytic performance of $\text{Pt}_2\text{Ir}_1/\text{CoP}$. It was found that the chemical and morphological characters of the loaded Ir in Ir/CoP (Figure R1), especially the loading (~ 1.0 wt.%) and size (~ 1.63 nm) were similar to those of Pt/CoP and $\text{Pt}_2\text{Ir}_1/\text{CoP}$ (Figure 3 and S11), therefore excluding their size influences on the catalytic performance.

Figure R1 | Characterizations of Ir/CoP. (a) TEM image and size distribution of Ir. (b) Inductively-coupled plasma mass spectrometry analysis of Ir/CoP.

The Ir/CoP catalysts showed an overpotential of 144 mV to reach $20 \text{ mA}/\text{cm}^2$ (η_{20}) and Tafel slope of 106.2 mV/dec (Figure R2), which were similar to these of bare CoP ($\eta_{20} = 156$ mV and Tafel slope = 108.1 mV/dec) as well as Pt/CoP ($\eta_{20} = 120$ mV and

Tafel slope = 103.1 mV/dec) and much higher than those of Pt₂Ir₁/CoP ($\eta_{20} = 7$ mV and Tafel slope = 25.2 mV/dec). The results indicated the non-dominant contributions of Ir itself in Pt₂Ir₁/CoP and the significance of the alloyed Pt₂Ir₁ for the improved catalytic activity.

Figure R2 | HER performance of bare CoP, Pt/CoP, Ir/CoP and Pt₂Ir₁/CoP catalysts in Ar-saturated 0.5 M H₂SO₄.

To support these results, the hydrogen adsorption and desorption behavior of Ir/CoP were further evaluated and compared with other control catalysts by the *operando* EIS and CV investigations. The recorded Nyquist plots were simulated by a double-parallel equivalent circuit model (Figure R3 and Table R1).

Figure R3 | Nyquist plots for Ir/CoP catalysts at various HER overpotentials. Zoom-in parts were correspondingly presented as inset. The scattered symbols represent the experimental results, and the solid lines are simulated fitting results. The inset also shows the equivalent circuit for the simulation. The fitted parameters are summarized in Table R1.

Table R1 | The fitted parameters of the EIS data of Ir/CoP for HER.

Catalysts	η [mV]	R_s [Ω]	T [F s ⁿ⁻¹]	R_1 [Ω]	n_1	R_2 [Ω]	C_ϕ [F]
Ir/CoP	0	3.52	0.0040	24.4	0.83	8886	0.0019
	-10	3.61	0.0042	24.1	0.91	6405	0.0025
	-20	3.62	0.0041	23.7	0.85	4766	0.0027
	-30	3.60	0.0043	23.4	0.86	3691	0.0031
	-40	3.62	0.0043	23.0	0.87	3002	0.0042
	-50	3.57	0.0046	22.9	0.85	2112	0.0071
	-60	3.58	0.0044	22.0	0.88	1466	0.0102
	-70	3.63	0.0041	21.7	0.82	999	0.0145
	-80	3.51	0.0040	21.4	0.85	740	0.0190
	-90	3.54	0.0045	20.8	0.86	566	0.0218
	-100	3.62	0.0043	20.2	0.88	406	0.0235
-110	3.60	0.0046	19.7	0.90	302	0.0245	

-120	3.58	0.0040	19.0	0.83	198	0.0255
-130	3.57	0.0039	18.1	0.86	110	0.0266
-140	3.56	0.0042	17.5	0.88	–	–

As shown in Figure R4a, the integration of C_ϕ vs. η profiles gives the hydrogen adsorption charge (Q_H) on catalyst surfaces during HER. The small Q_H value of Ir on Ir/CoP ($Q_H[\text{Ir/CoP}] - Q_H[\text{CoP}] = 188 \mu\text{C}$) support that the enhancement from Ir itself for Pt₂Ir₁/CoP catalysts was too slight to dominate their hydrogen adsorption behavior.

Figure R4 | (a) Plots of C_ϕ vs. η for bare CoP, Pt/CoP, Ir/CoP, Pt₂Ir₁/CoP, Pt/rGO, Pt₂Ir₁/rGO, SCN-Pt/CoP and Pt₂Ir₁/CoP during HER in Ar-saturated 0.5 M H₂SO₄. (b) CV of the bare CoP, Pt/CoP, Ir/CoP, Pt₂Ir₁/CoP, Pt/rGO, Pt₂Ir₁/rGO, SCN-Pt/CoP and SCN-Pt₂Ir₁/CoP catalysts in 0.5 M H₂SO₄ with a scan rate of 50 mV/s.

To investigate the hydrogen desorption behavior, *operando* CV investigations were also carried out on Ir/CoP and other control catalysts and their hydrogen desorption peak was monitored during CV scanning in the double layer region. As shown in Figure R4b, the CV curves showed that the intensity of the hydrogen desorption peaks for bare CoP, Pt/CoP and Ir/CoP was equally weak, supported that the enhancement from Ir itself for Pt₂Ir₁/CoP catalysts was too weak to dominate their hydrogen desorption behavior.

Overall, the above facts further confirm that Ir itself in Pt₂Ir₁/CoP catalysts should not dominate their HER activity improvement. **For a better understanding, the updated discussions can be found in our revised manuscript (Page 7-16).**

Reviewer 3

The revision is OK but the Response is so excessively long (43 pages, not to forget also 39 pages of the paper) that it reminds the “filibuster method” in some parliamentary proceedings. Anyways, the authors tried hard to address reviewers’ concerns. I still do not see clear convincing evidence of the spillover process but I can accept it here as an overall plausible scenario of the process, OK. I do see and sympathize with the stronger reservations of the Reviewer 1. Yet from my perspective the paper overall is thorough and its publication can be useful, even though leaving some moot points debatable in the future. I can endorse its acceptance to Nat. Comm. at this point.

Response: We thank the reviewer’s constructive comments on our manuscript. Through the revision, we feel a significant improvement in the quality of this manuscript. In the future, we expect to directly observe the electrocatalytic hydrogen spillover phenomenon based on the development of methodology and technology and further provides the insights on the hydrogen-spillover-based HER electrocatalysts.